# Dynamic control of proinflammatory cytokines Il-1β and Tnf-α by macrophages in zebrafish spinal cord regeneration

Themistoklis M. Tsarouchas[1], Daniel Wehner [1,4], Leonardo Cavone[1], Tahimina Munir[1], Marcus Keatinge[1], Marvin Lambertus[1,5], Anna Underhill[1], Thomas Barrett [1], Elias Kassapis [1], Nikolay Ogryzko [2], Yi Feng [2], Tjakko J. van Ham [3], Thomas Becker[1] & Catherina G. Becker [1]

Spinal cord injury leads to a massive response of innate immune cells in non-regenerating mammals, but also in successfully regenerating zebrafish. However, the role of the immune response in successful regeneration is poorly defined. Here we show that inhibiting inflammation reduces and promoting it accelerates axonal regeneration in spinal-lesioned zebrafish larvae. Mutant analyses show that peripheral macrophages, but not neutrophils or microglia, are necessary for repair. Macrophage-less *irf8* mutants show prolonged inflammation with elevated levels of Tnf-α and Il-1β. Inhibiting Tnf-α does not rescue axonal growth in *irf8* mutants, but impairs it in wildtype animals, indicating a pro-regenerative role of Tnf-α. In contrast, decreasing Il-1β levels or number of Il-1β+ neutrophils rescue functional regeneration in *irf8* mutants. However, during early regeneration, interference with Il-1β function impairs regeneration in *irf8* and wildtype animals. Hence, inflammation is dynamically controlled by macrophages to promote functional spinal cord regeneration in zebrafish.

[1] Centre for Discovery Brain Sciences, University of Edinburgh, The Chancellor's Building, 49 Little France Crescent, Edinburgh EH16 4SB, UK. [2] MRC Centre for Inflammation Research, Queen's Medical Research Institute, University of Edinburgh, Edinburgh EH16 4TJ, UK. [3] Department of Clinical Genetics, Erasmus University Medical Center, Wytemaweg 80, 3015 CN Rotterdam, The Netherlands. [4] Present address: Technische Universität Dresden, DFG-Center of Regenerative Therapies Dresden, Fetscherstraße 105, Dresden 01307, Germany. [5] Present address: Department of Pharmaceutical Biosciences, School of Pharmacy, University of Oslo, 0316 Oslo, Norway. These authors contributed equally: Thomas Becker, Catherina G. Becker. Correspondence and requests for materials should be addressed to T.B. (email: thomas.becker@ed.ac.uk) or to C.G.B. (email: catherina.becker@ed.ac.uk)

Zebrafish, in contrast to mammals, are capable of functional spinal cord regeneration after injury. Recovery of swimming function critically depends on regeneration of axonal connections across the complex non-neural injury site[1,2]. It is therefore important to determine the factors that allow axons to cross the lesion site in zebrafish.

In mammals, a prolonged immune response, consisting of pro-inflammatory macrophages[3], microglia cells[4] and neutrophils[5] together with cytokines released from other cell types, such as endothelial cells, oligodendrocytes, or fibroblasts[6] contribute to an inhibitory environment for axonal regeneration. However, activated macrophages can also promote axonal regeneration[7–9], suggesting complex roles of the immune response after spinal injury.

In zebrafish, we can dissect the roles of these cell types in successful functional spinal cord repair[10]. Zebrafish possess an innate immune system from early larval stages and develop an adaptive immune system at juvenile stages, similar to those in mammals[11]. Indeed, microglia is activated after spinal cord injury in adult[12,13] and larval zebrafish[14], suggesting functions of innate immune cells in repair. Adaptive immunity is also important for spinal cord regeneration[15].

Larval zebrafish regenerate more rapidly than adults. Axonal and functional regeneration is observed within 48 h after spinal cord injury in 3 day-old larvae[1,2]. At the same time, the larval system presents complex tissue interactions that allow us to analyse how axons cross a non-neural lesion environment. For example, axons encounter Pdgfrb+ fibroblast-like cells that deposit regeneration-promoting Col XII in the lesion site in a Wnt-signalling dependent manner[1]. These cells and molecules are present also in the injury sites of adult zebrafish and mammals[1,6]. How immune cells contribute to this growth-conducive lesion site environment in zebrafish is unclear.

Here we show that peripheral macrophages control axonal regeneration by producing pro-regenerative tumour necrosis factor alpha (Tnf-α) and by reducing levels of interleukin-1 β (Il-1β). While early expression of *il-1β* promotes axonal regeneration, prolonged high levels of Il-1β in the macrophage-less *irf8* mutant are detrimental. Preventing formation of Il-1β producing neutrophils or inhibiting excess *il-1β* directly, largely restored repair in *irf8* mutants. This indicates that regulation of a single immune system-derived factor, Il-1β, is a major determinant of successful spinal cord regeneration.

## Results

### The immune response coincides with axonal regeneration. 
We analysed axonal regrowth in larval zebrafish that underwent complete spinal cord transection at 3 days post-fertilisation (dpf) in relation to invasion of the injury site by different cell types. Axons were present in the injury site by 1 day post-lesion (dpl). The thickness of the axonal bundle that connects the injured spinal cord increased up to 2 dpl and thereafter plateaued for up to at least 4 dpl (Supplementary Fig. 1A). The thickness of the connecting axon bundle positively correlated with the recovery of touch-evoked swimming distance for individual animals at 2 dpl (Supplementary Fig. 2A–C). This is consistent with previous results showing continuous axon labelling over the lesion site (axon bridging) in 80% of animals by 2 dpl, which then plateaued. Presence of an axon bridge correlates with functional recovery, as animals without axon bridge showed worse recovery of touch-evoked swimming distance[1] and re-lesioning abolished functional recovery[14]. Hence, a percent score of larvae with bridged injury sites is a quick and reliable measure for anatomical repair[1,14].

After injury, we observed a rapid and massive influx of immune cells, with neutrophils (Mpx+) peaking at 2 h post-lesion (hpl) and macrophages (*mpeg1*:GFP+; 4C4-) and microglia (*mpeg1*:GFP+; 4C4+) accumulating in the lesion site a few hours later and peaking at 2 dpl (Fig. 1a, Supplementary Movie 1). Myelinating cells (*cldnK*:GFP+) and endothelial cells (*fli1*:GFP+) were not abundant in the lesion site during axonal regrowth (Supplementary Fig. 1C, D), in contrast to functionally important *pdgfrb*:GFP+ fibroblasts[1] that were present in the lesion site at 1 dpl, peaking at 2 dpl (Supplementary Fig. 1B). This suggests that myelinating cells and endothelial cells are not essential for axon bridging. However, at later time points after injury, axons were clearly associated with processes of myelinating cells (Supplementary Fig. 1C), which may impact functional repair. In contrast, the spatio-temporal pattern of immune cell invasion of the injury site suggests an early role for the immune system in orchestrating axon growth over the lesion site.

### Immune system activation promotes axonal regeneration. 
To determine the importance of the immune reaction, we inhibited it using the anti-inflammatory synthetic corticosteroid dexamethasone[14]. This reduced the number of microglia[14], macrophages and neutrophils in the injury site (Fig. 1b–d) and the proportion of larvae exhibiting axon bridging (control: 78% of examined animals, dexamethasone: 30%; Fig. 1e). The average thickness of axon bridges was also reduced by dexamethasone treatment and correlated with impaired recovery of touch-evoked swimming distance. (Supplementary Fig. 2A–C). *gfap*:GFP+ astroglia-like processes that cross the injury site slightly later than axons[1] also showed reduced bridging, from 77.6% of examined animals to 48.3% (Supplementary Fig. 2D) under dexamethasone treatment. In addition, depleting the number of immune cells with a well-established morpholino combination against *pu.1* and *gcsfr*[16] reduced the proportion of larvae with axonal bridges from 81% of examined animals to 57% (Supplementary Fig. 3A, B).

For a gain-of-function approach, we used incubation with bacterial lipopolysaccharides (LPS)[17]. This increased the number of neutrophils and macrophages in the lesion site (Fig. 1f–h). To detect a potential accelerating effect on axonal regrowth, we analysed larvae at 18 hpl, when axonal regeneration was incomplete in untreated animals. This showed an increase in the proportion of larvae with axonal bridges from 41% of examined animals in wildtype to 60% in LPS-treated animals (Fig. 1i). Hence, immune system activity is necessary for and promotes axonal regeneration across a spinal lesion site.

### Macrophages determine regenerative success. 
To analyse the role of different immune cell types in repair, we used mutants. In mutants for the macrophage-lineage determining transcription factor *irf8*, macrophages and microglial cells, but not neutrophils are missing during early development[18]. Homozygous mutants are adult viable and show no overt developmental aberrations, except for an increased number of neutrophils[18]. In situ hybridisation for the macrophage and microglia marker *mpeg1* confirmed expression in the ventral trunk of unlesioned larvae and in a spinal lesion site at 2 dpl in wildtype larvae, but complete absence of signal in unlesioned and lesioned *irf8* larvae (Fig. 2a).

Next, we determined axonal regrowth and recovery of parameters of swimming capacity in *irf8* mutants compared to wildtype animals. Wildtype and mutants showed comparable proportions of animals with axon bridges at 1 dpl (wildtype: 44% of examined animals; *irf8* mutant: 43%). At 2 dpl, however, axonal continuity was observed in 80% of wildtype animals but only in 41% of *irf8* mutants (Fig. 2b). At 5 dpl–2.5 times as long as wildtype animals need for maximal axon bridging—the proportion of mutant larvae with bridged lesion sites was increased compared to 2 dpl (55% of examined animals vs. 41%), but

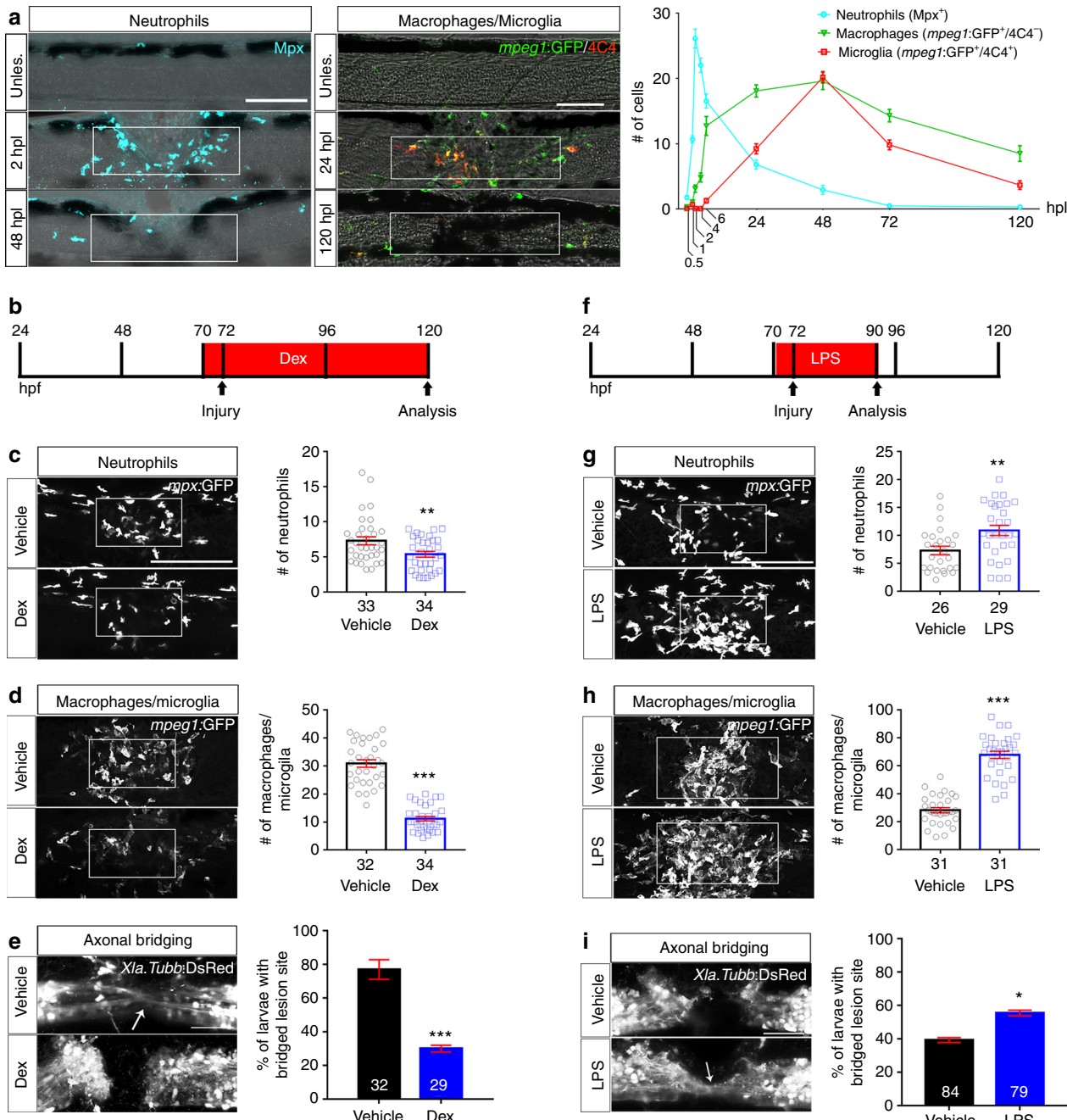

**Fig. 1** Spinal injury leads to an inflammatory response that promotes axonal regeneration. **a** Neutrophils, macrophages, and microglial cells show different dynamics after injury. Neutrophils (Mpx⁺) accumulate in the injury site very early, peaking at 2 hpl. Macrophages (*mpeg1*:GFP⁺/4C4⁻) and microglial cell (*mpeg1*:GFP⁺/4C4⁺) numbers peak at 48 hpl. Fluorescence images were projected onto transmitted light images. **b**–**e** Incubation with dexamethasone (timeline in **b**) reduces neutrophil and macrophage numbers (**c**, **d**; Mann–Whitney *U*-test: **P < 0.01, ***P < 0.001), as well as the proportion of animals with axonal bridging (**e**; Fisher's exact test: ***P < 0.001). **f**–**i**, Incubation of animals with LPS during early regeneration (timeline in **f**) increased numbers of neutrophils and macrophages (**g**, **h**; *t*-test: **P < 0.01, ***P < 0.001), as well as the proportion of animals with axonal bridging at 24 hpl (**i** Fisher's exact test: *P < 0.05). Lateral views of the injury site are shown; rostral is left. Rectangles indicate region of quantification; arrows indicate axonal bridging. Scale bars: 50 μm; Error bars indicate SEM

regenerative success was still strongly reduced compared to wildtype controls (55% of examined animals vs. 87%; Fig. 2c).

Analysing touch-evoked swimming, we found that wildtype animals swam comparable distances to unlesioned controls at 2 dpl, as previously described[1]. In contrast, recovery of touch-evoked swimming distance in *irf8* larvae plateaued at 2 dpl and did not reach levels of unlesioned animals to at least 5 dpl

(Fig. 2c). This indicates that in the absence of macrophages and microglia in *irf8* mutants, initial axonal regeneration is unaffected, but axonal regrowth and functional recovery after spinal cord injury are impaired long-term.

To determine the importance of microglia for regeneration, we analysed *csf1ra/b* double-mutants (see Methods) in which the function of colony-stimulating factor 1 receptor (Csf1r) is

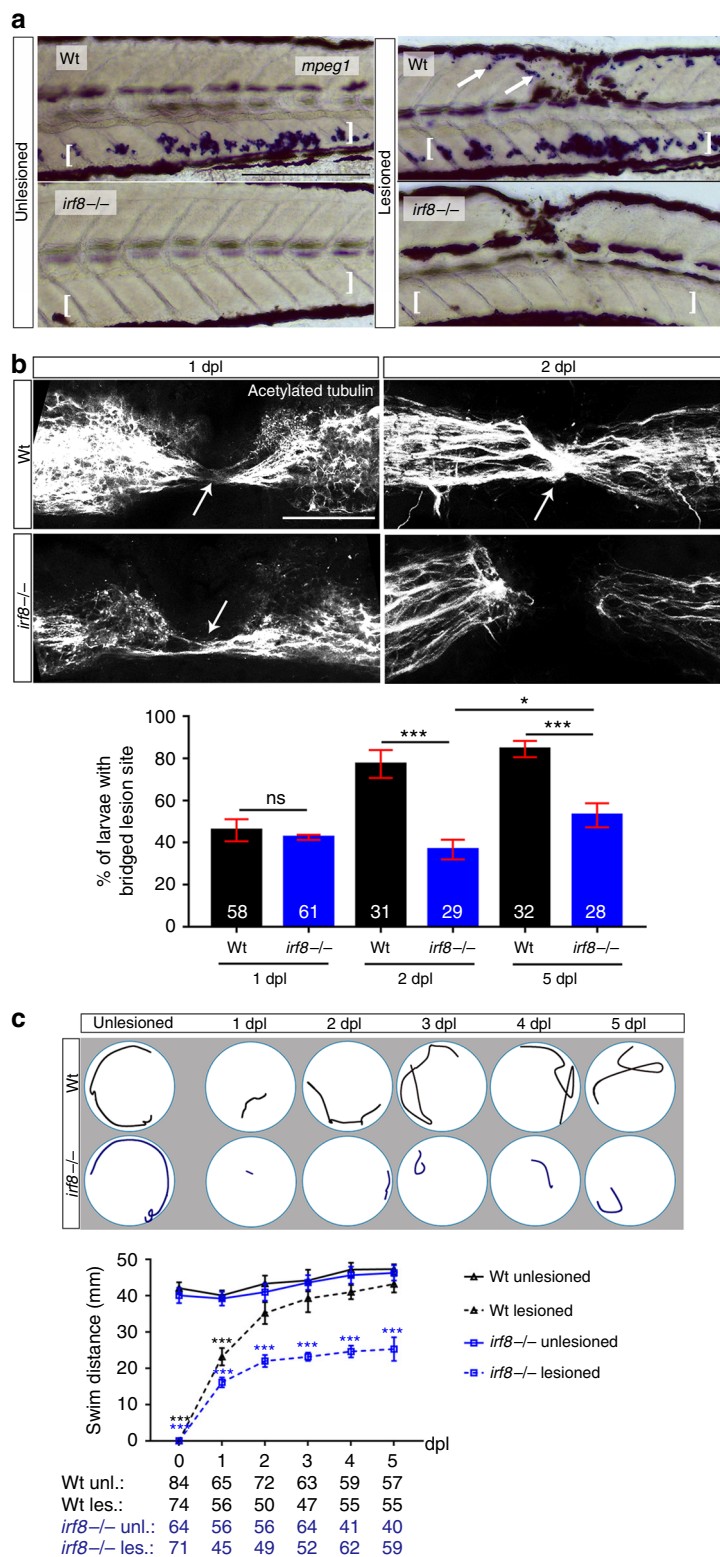

compromised. Csf1r is selectively needed for microglia differentiation[19]. After injury in *csf1ra/b* mutants, we observed a strong reduction in the number of microglial cells (to 17% of wildtype), but an increase in macrophage numbers (by 55% compared to wildtype) in the injury site (Fig. 3a, c). Interestingly, neutrophil numbers were also strongly reduced (to 64.1% of wildtype at 2 hpl and 16% at 1 dpl) (Fig. 3b), perhaps due to feedback regulation from increased macrophage numbers. Whereas microglia cells

were reduced in number in the entire fish, neutrophils were still present in the ventral trunk area. In these mutants, axon bridging was unimpaired (Fig. 3d). Hence, microglia are not necessary for axonal regeneration and reduced numbers of neutrophils do not negatively affect axonal regrowth. Combined with results from the *irf8* mutant, this indicates that recruitment of peripheral macrophages is critical for successful spinal cord regeneration.

**Fig. 2** In the *irf8* mutant, axonal regeneration and functional recovery after injury show long-term impairment. **a** In situ hybridisation for *mpeg1* confirms the absence of macrophages and microglial cells before and after injury in the *irf8* mutant compared to controls. Arrows indicate labelling around the injury site and brackets indicate the ventral area of the larvae where the macrophages can be found in the circulation. Note that blackish colour is due to melanocytes. **b** Quantification of the proportion of larvae with axonal bridging (anti-acetylated Tubulin) shows that at 1 dpl, axonal bridging is unimpaired in *irf8* mutants, whereas at 2 dpl, *irf8* mutants fail to show full regrowth and even by 5 dpl, the proportion of *irf8* larvae with a bridged lesion site is still lower than in wildtype controls (Fisher's exact test: ***$P < 0.001$, n.s. indicated no significance). **c** *Irf8* mutants never fully recover touch-evoked swimming distance in the observation period, whereas wildtype control animals do. Representative swim tracks are displayed. Note that unlesioned *irf8* larvae show swimming distances that are comparable to those in wildtype controls (Two-way ANOVA: $F_{15,1372} = 11.42$, $P < 0.001$; unles. = unlesioned, les. = lesioned). All lesions are done at 3 dpf. Lateral views of the injury site are shown; rostral is left. Arrows indicate axonal bridging. Scale bars: 200 μm in **a** and 50 μm in **c**. Error bars indicate SEM

**Macrophages are not necessary for Col XII deposition.** Next, we asked whether macrophages act via a previously reported regeneration-promoting mechanism, comprising Wnt-dependent deposition of Col XII in the lesion site by *pdgfrb*:GFP+ fibroblast-like cells[1]. Inhibition of the immune response with dexamethasone did not inhibit appearance of *pdgfrb*:GFP+ fibroblast-like cells in the lesion site (Supplementary Fig. 4A). By crossing a reporter line for Wnt pathway activity (*6xTCF*:dGFP)[1] into the *irf8* mutant, we found that activation of the pathway was unaltered in the mutant (Supplementary Fig. 4E). Similarly, expression of *col12a1a* and *col12a1b* mRNA in *irf8* mutants was indistinguishable from that in wildtype animals (Supplementary Fig. 4B). Deposition of Col I protein and mRNA expression of 11 other ECM components were also not altered in the *irf8* mutant at 1 and 2 dpl (Supplementary Fig. 4B, C). Moreover, immuno-labelling against Tp63 showed that by 2 dpl, the injury site in the *irf8* mutants was completely covered by basal keratinocytes, an additional source of Col XII[1], as in wildtype animals (Supplementary Fig. 4D). In contrast, a PCR screen of 21 potentially macrophage-derived ECM-modifying matrix metalloproteinases[20] (*mmps*) indicated lower mRNA levels for *mmp11a*, *mmp16a/b*, *mmp24*, and *mmp28* in the injury site of *irf8* mutants compared to wildtype animals (Supplementary Fig. 5A, B). This suggests a potential for macrophages to alter the lesion site ECM with Mmps. Overall, macrophages do not overtly regulate Wnt-signalling, deposition of some crucial ECM components or basal keratinocyte coverage of the injury site during regeneration.

**Cellular debris is not a major impediment to regeneration.** Macrophages could serve as a substrate for axon growth or promote regeneration by removing debris by phagocytosis—a major function of macrophages in peripheral nerve regeneration in zebrafish[21,22]. In time-lapse movies of double transgenic animals in which neurons (*Xla.Tubb*:DsRed) and macrophages (*mpeg1*:GFP) were labelled (Supplementary Fig. 6B and Supplementary Movie 2) we observed axons crossing the spinal lesion site at the same time macrophages migrated in and out of the injury site. However, no obvious physical interactions between these cell types were observed, making it unlikely that macrophages acted as an axon growth substrate.

We frequently observed macrophages ingesting neuronal material and transporting that away from the injury site in time-lapse movies (Supplementary Fig. 6B and Supplementary Movie 2). In agreement with this observation, TUNEL labelling indicated strongly increased levels of dead or dying cells in the late (48 hpl), but not the early (24 hpl) phase of axonal regeneration in the injury site of *irf8* mutants (Supplementary Fig. 6A).

To determine the impact of debris on regeneration, we prevented cell death and consequently debris accumulation in *irf8* larvae using the pan-caspase inhibitor QVD[23], that is functional in zebrafish[24]. This treatment led to lower debris levels that were comparable to those seen in wildtype larvae at 2

dpl (Supplementary Fig. 6C), but failed to increase regenerative success in *irf8* mutants (control, 38% of examined larvae with axon bridges; QVD, 40%. Thickness of axon bridge: control 19.03 +/−2.14 μm; QVD: 18.32+/−2.53 μm; *t*-test: $P > 0.05$. Supplementary Fig. 6E). Conversely, inhibiting debris phagocytosis with the pharmacological inhibitor O-phospho-L-serine (L-SOP)[25] in wildtype animals increased levels of debris in the injury site, but did not impair axonal bridging (Supplementary Fig. 7A–C). This suggests no obvious connection between debris levels and/or phagocytosis and regenerative success.

**Pro-and anti-inflammatory phases are altered in *irf8* mutants.** To determine a possible role of cytokines in the regenerative failure of *irf8* mutants, we analysed relative levels of pro-and anti-inflammatory cytokines in the lesion site during regeneration in wildtype animals and *irf8* mutants by qRT-PCR. In wildtype animals, expression levels of pro-inflammatory cytokines *il-1β* and *tnf-α* were high during initial regeneration (>25-fold for *il-1β*; >12-fold for *tnf-α* for approximately to 12 hpl) and reduced again during late regeneration (12–48 hpl), although still elevated compared to unlesioned controls (Fig. 4a, b). Anti-inflammatory cytokines, such as *tgf-β1a* and *tgf-β3* were expressed at low levels during initial regeneration, and upregulated during late regeneration (approximately 3-fold for *tgf-β1a* and 2-fold for *tgf-β3*), indicating a bi-phasic immune response within the 48-h time frame of analysis (Fig. 4c, d).

In *irf8* mutants, levels of pro-inflammatory cytokines remained high during the late phase of regeneration (Fig. 4a, b) and anti-inflammatory cytokines were not upregulated (Fig. 4c, d), resulting in a sustained pro-inflammatory environment in *irf8* mutants.

The lack of anti-inflammatory cytokines correlated with the lack of macrophages and microglia in *irf8* mutants. We performed qRT-PCR in fluorescence activated flow sorted *mpeg1*:GFP cells in wildtype animals to determine whether macrophages and microglial cells expressed *tgf-β1a and tgf-β3*. *mpeg1*:GFP+ cells, but also *mpeg1*:GFP− cells expressed these cytokines (Supplementary Fig. 8A). In situ hybridisation showed wide-spread labelling with some more strongly labelled cells around the injury site in wildtype, but not *irf8* mutants (Supplementary Fig. 8B). This is consistent with expression of *tgf-β1a* and *tgf-β3* in microglia/macrophages and other cell types[26]. Hence, the immune response is bi-phasic with an initial pro-inflammatory phase, followed by an anti-inflammatory phase in wildtype animals. In the absence of macrophages in *irf8* mutants, animals fail to switch to an anti-inflammatory state.

**Tnf-α promotes axonal regeneration.** To determine whether increased levels of pro-inflammatory cytokines contributed to impaired axon growth in *irf8* mutants, we first inhibited Tnf-α signalling. Pomalidomide, a pharmacological inhibitor of Tnf-α release[27], had no effect on axonal regrowth in *irf8* mutants. In

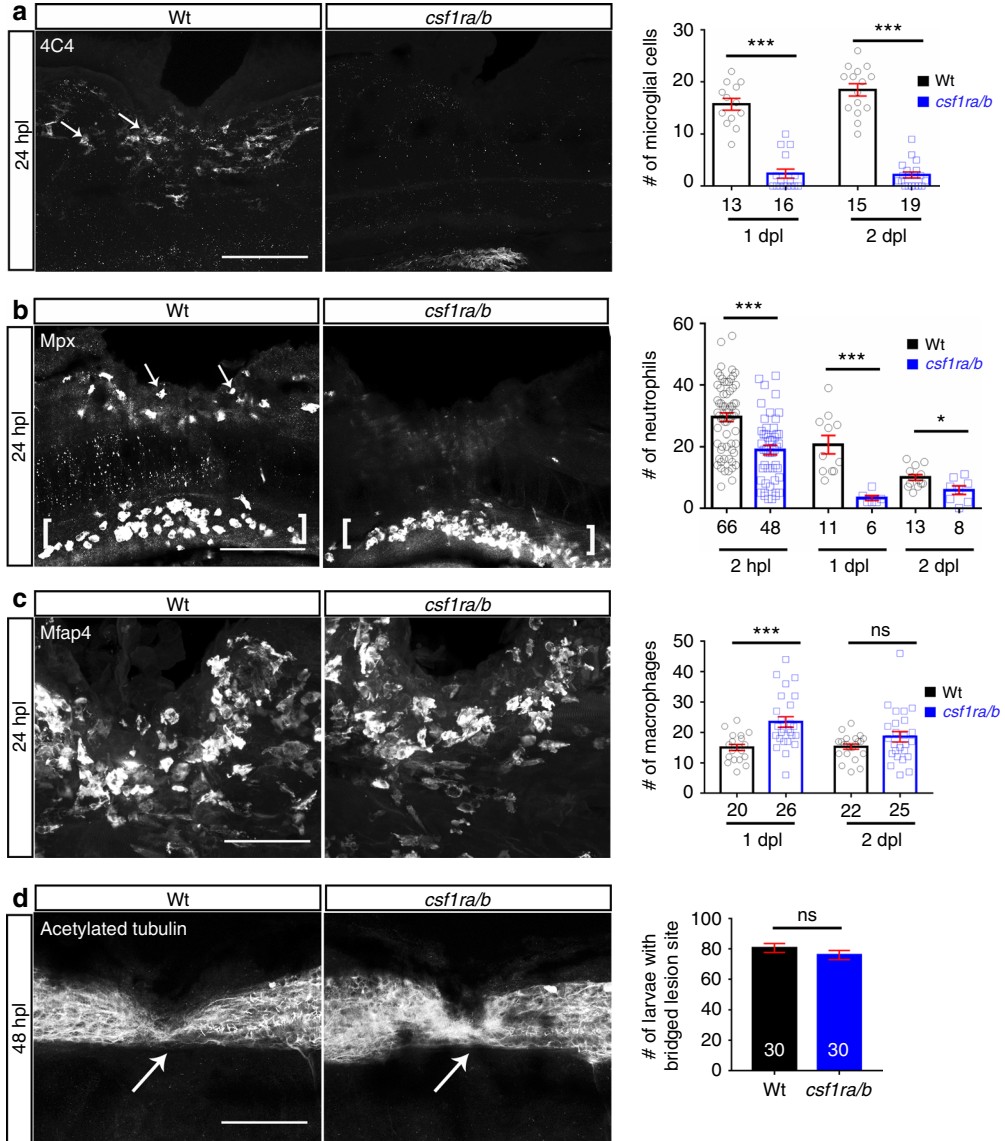

**Fig. 3** Absence of microglial cells and reduced neutrophil numbers do not affect axon bridging. **a** Numbers of microglial cells (4C4+; arrows) in the injury site of the *csf1ra/b* mutants are much lower than in wildtype animals (*t*-test: ***P < 0.001). **b** Fewer neutrophils (Mpx+) are found in the injury site (arrows) of *csf1ra/b* mutants than in wildtype animals (*t*-test: *P < 0.05, ***P < 0.001). Note neutrophils ventral to the injury site (brackets). **c** The number of macrophages (Mfap4+) is increased in the injury site in the mutants at 1 dpl, but not at 2 dpl (*t*-test: ***P < 0.001, ns indicates no significance). **d** Immunostaining against acetylated tubulin shows that axon bridging (arrows) is not affected in the mutants compared to wildtype animals at 2 dpl (Fisher's exact test: ns indicates no significance). Lateral views of the injury site are shown; rostral is left. Wt = wildtype; Scale bars: 50 μm in **a**, **b**, **d**; 25 μm in **b**. Error bars indicate SEM

contrast, in wildtype animals Pomalidomide strongly inhibited axon bridging at 1 dpl (control: 62% of examined animals showed an axonal bridge; Pomalidomide: 36%) and 2 dpl (control: 75% of examined animals; Pomalidomide: 45%) (Fig. 5a).

To confirm pharmacological results, we targeted *tnf-α* by using CRISPR manipulation with a gene-specific guideRNA (gRNA). Injection of the gRNA into the zygote efficiently mutated the gene as shown by restriction fragment length polymorphism (RFLP) analysis (Fig. 5b) and produced function-disrupting insertion-deletion mutations in a highly conserved domain[28] (Supplementary Table 1). Western blots of 4-day old larvae and immunohistochemistry in the injury site showed robustly reduced Tnf-α protein levels (Supplementary Fig. 9A, B).

Axon bridging was inhibited in wildtype animals by *tnf-α* gRNA injection in a way that was comparable to drug treatment (1 dpl: control: 51% of examined animals showed bridging; gRNA: 27%; 2 dpl: control: 88% of examined animals showed bridging; gRNA: 40%). At 5 dpl, axon bridging was still strongly impaired (control: 84.1% of examined animals showed bridging; gRNA: 38.3%; Fig. 5c), indicating long-term impairment of regeneration. Hence, *tnf-α* dysregulation is not a major cause of regenerative failure in *irf8* mutants, but *tnf-α* is necessary for axonal regeneration in wildtype animals.

To determine which cells expressed *tnf-α* in wildtype animals, we used immunohistochemistry for L-Plastin, labelling all immune cells, in *tnf-α*:GFP transgenic fish (Fig. 6d). Nearly all *tnf-α*:GFP+ cells co-labelled with L-Plastin (96%) at 12 hpl. Thus, expression of *tnf-α* occurred mainly in immune cells (Fig. 6a). Double-labelling *tnf-α*:GFP reporter fish with neutrophil (Mpx), microglia (4C4) and macrophage (Mfap4) markers at 24 hpl, when axons were actively growing, indicated that >95% of *tnf-α*:GFP+ cells in the injury were peripheral macrophages. However

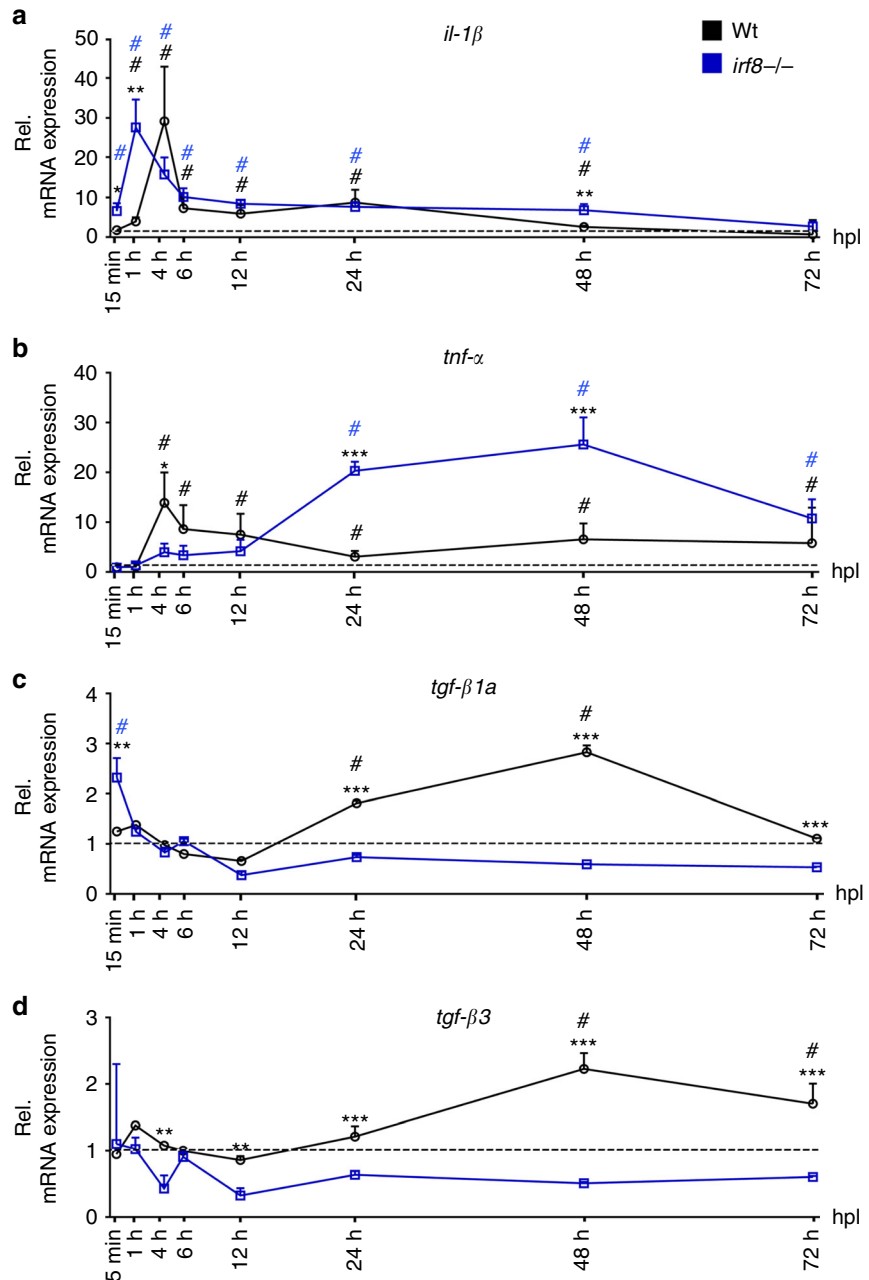

**Fig. 4** Inflammation is bi-phasic and dysregulated in *irf8* mutants. **a**, **b** Absence of macrophages in the *irf8* mutant fish leads to increased *il-1β* and *tnf-α* mRNA levels during the late stage of inflammation (>12 hpl). An early peak in *tnf-α* expression is missing in *irf8* mutants. **c**, **d** Expression of anti-inflammatory cytokines, *tgf-β1a* and *tgf-β3*, which peak during late regenerative phases in wildtype animals, is strongly reduced in *irf8* mutants (*t*-tests: \**P* < 0.05, \*\**P* < 0.01, \*\*\**P* < 0.001; wt = wildtype animals). # indicates statistical significance when compared to unlesioned animals. Error bars indicate SEM

other cell types, such as neurons[29], may also express *tnf-α*. More than 72% of macrophages were *tnf-α*:GFP[+], whereas for microglia (<6.5%) and neutrophils (<0.7%) the proportion was much smaller. Over time, the proportion of *tnf-α*:GFP[+] macrophages was reduced (from 72.5% at 1 dpl to 59% at 2 dpl) (Fig. 6b). Our observations suggest that macrophages promote regeneration by expressing *tnf-α*.

To elucidate effects of *tnf-α* inhibition, we determined numbers of neutrophils and macrophages/microglia at 24 and 48 hpl in *tnf-α* gRNA injected animals. This showed no changes in macrophages, but a 49.7% increase in the number of neutrophils at 1 dpl (Fig. 6c). qRT-PCR indicated that *il-1β* mRNA levels were increased by 108%, whereas *tnf-α*, *tgf-β1a* and *tgf-β3* mRNA levels remained

unchanged at 2 dpl (Fig. 6d). This suggests a moderate enhancement of the pro-inflammatory response when Tnf-α is inhibited.

**Il-1β inhibits regeneration in *irf8* mutants.** To test whether sustained high levels of Il-1β were responsible for regenerative failure in *irf8* mutants, we interfered with *il-1β* function in three different ways. Firstly, we inhibited caspase-1, which is necessary for activation of Il-1β, using the pharmacological inhibitor YVAD that is functional in zebrafish[30] (Fig.7a–e). Secondly, we disrupted *il-1β* RNA splicing with an established morpholino (Supplementary Fig. 10A–D)[31]. Finally, we targeted *il-1β* in a CRISPR approach (Supplementary Fig. 10E–H).

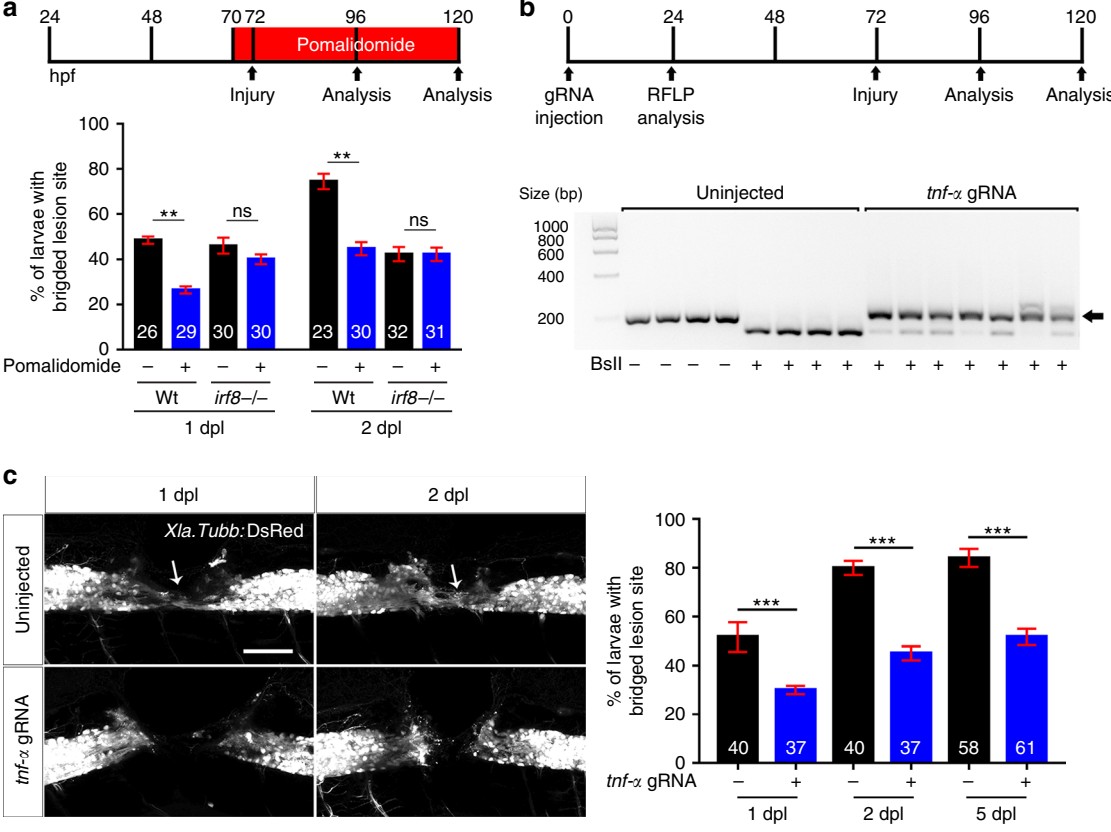

**Fig. 5** Tnf-α is essential for axonal regeneration. **a** Tnf-α inhibition by Pomalidomide reduces the proportion of wildtype animals with axon bridging at 1 and 2 dpl. No effect is observed in *irf8* mutants (Two-way ANOVA followed by Bonferroni post-test: $F_{3,16} = 12.16$, **$P < 0.01$, n.s indicates no significance). **b** CRISPR/Cas9-mediated disruption of *tnf-α* is effective as shown by RFLP analysis. This reveals efficient somatic mutation in the gRNA target site, indicated by resistance to restriction endonuclease digestion (arrow). **c** Axonal bridging (arrow; *Xla.Tubb*:DsRed⁺) is strongly impaired after disruption of the *tnf-α* gene. (Fisher's exact test: ***$P < 0.001$) and the impairment persists at 5 dpl. Lateral views of the injury site are shown; rostral is left. Scale bar: 50 μm. Error bars indicate SEM

To determine whether interfering with Il-1β function mitigated inflammation in *irf8* mutants, we quantified immune cells, expression of *il-1β*, *tnf-α*, and dead cells. Indeed, after YVAD treatment we observed a reduction of neutrophil peak numbers (by 38% at 2 hpl; Fig. 7b), as well as strongly reduced levels of *il-1β* and *tnf-α* mRNA expression (at 2 dpl; Fig. 7a) in *irf8* mutants. Moreover, the number of TUNEL⁺ cells was reduced at 2 dpl in the *irf8* mutant, but not to wildtype levels (Fig. 7c). In lesioned wildtype animals, YVAD reduced peak numbers of neutrophils (by 40% at 2 hpl) and macrophages (by 28% at 48 hpl), but no influence on low numbers of TUNEL⁺ cells at 2 dpl was observed. Hence, interfering with Il-1β function reduces inflammation in *irf8* mutants and wildtype animals.

Axon bridging in wildtype animals was not affected by YVAD treatment at 2 dpl (control: 79% of examined animals showed bridging; YVAD: 78%) (Fig. 7d), indicating that high levels of Il-1β were not necessary for axonal regeneration. In contrast, in YVAD-treated *irf8* mutants, we observed a remarkable rescue of axon bridging at 2 dpl (control: 38% of examined animals showed bridging; YVAD: 69%) (Fig. 7d).

Injecting a well-established[31] morpholino targeting *il-1β* into *irf8* mutants at the one-cell-stage inhibited *il-1β* splicing (Supplementary Fig. 10A, D). Morpholino-injected animals showed a rescue of axon bridging at 2 dpl (control: 40% of examined animals showed bridging; YVAD: 60%) (Supplementary Fig. 10B, C).

Finally, injecting a gRNA targeting *il-1β* at the one-cell stage led to somatic mutation in the target site of *il-1β*, indicated by

RFLP analysis (Supplementary Fig. 10E, H). This strongly rescued axonal bridging in lesioned *irf8* mutants (control: 40% of examined animals showed bridging; acute *il-1β* gRNA: 70%) (Supplementary Fig. 10F, G). Hence, three independent manipulations show that excessive *il-1β* levels in *irf8* mutants are a key reason for impaired axonal regeneration.

**Il-1β promotes axonal regeneration during the early regeneration**. To determine whether roles of Il-1β and general inflammation differed for different phases of the inflammation, we separately analysed early (0–1 dpl; Supplementary Fig. 11A) and late (1–2 dpl; Supplementary Fig. 11B) regeneration by drug incubation. During the early phase, YVAD treatment led to a weak inhibition of axonal regeneration in both wildtype (control: 58% of examined animals showed bridging; YVAD: 41%) and *irf8* mutants (control: 41% of examined animals showed bridging; YVAD: 36%). Similarly, dexamethasone treatment inhibited axonal regeneration in both wildtype (control: 57.5% of examined animals showed bridging; dexamethasone: 36.6%) and *irf8* (control: 44.6% of examined animals showed bridging; dexamethasone: 34%). Interestingly, while LPS promoted regeneration in the early phase in wildtype animals (control: 53.2% of examined animals showed bridging; LPS: 68.7%), it was detrimental in *irf8* mutants (control: 39% of examined animals showed bridging; LPS: 25.5%), perhaps because baseline inflammation was already high in the mutant.

During late regeneration, only dexamethasone had an inhibitory effect in wildtype animals (from 82.1% of examined animals that showed bridging to 64.4%). YVAD had no effect in

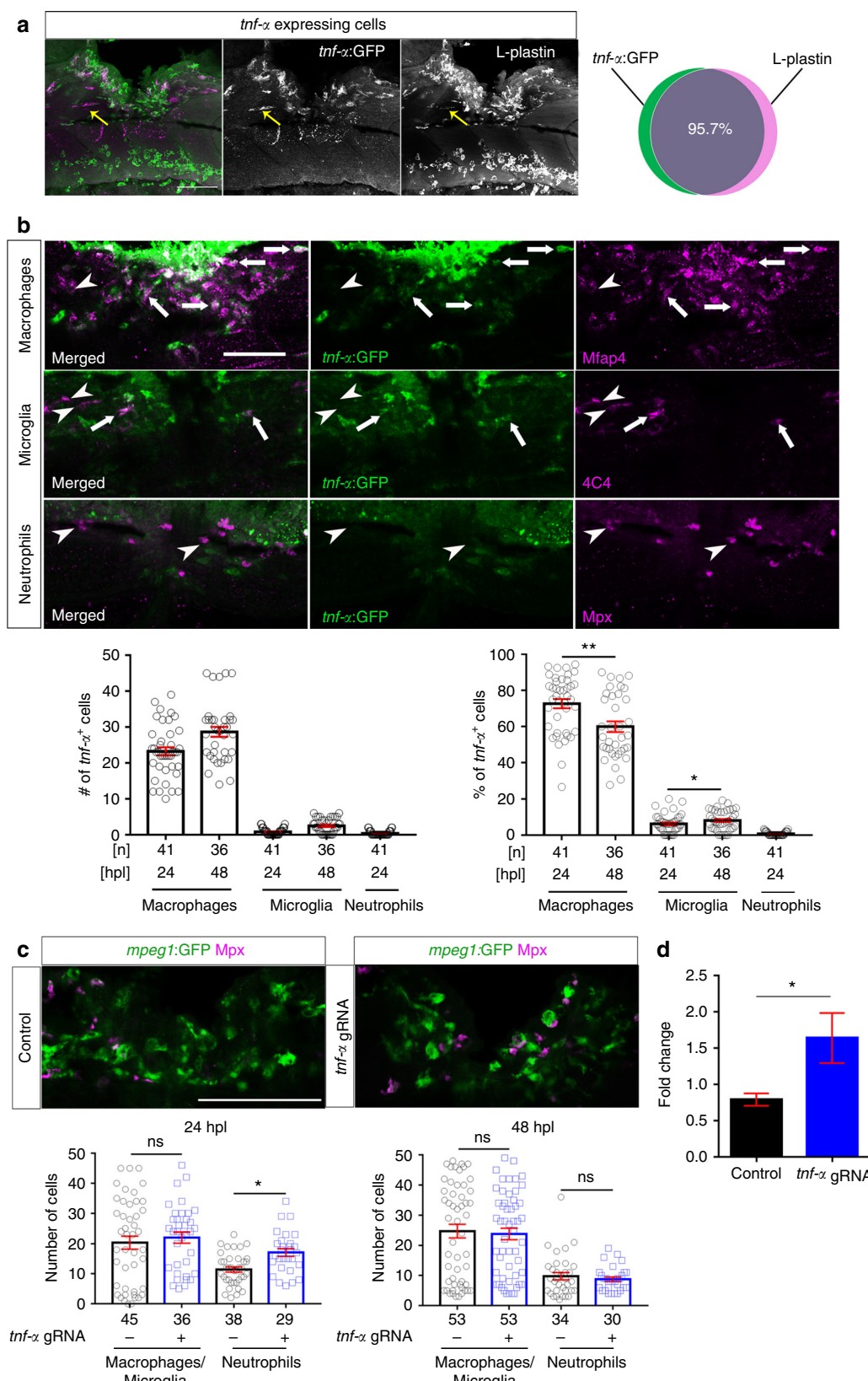

wildtype animals during late regeneration, when *il-1β* was already down-regulated (control: 81% crossing; YVAD: 78% crossing), but a strong rescue effect of YVAD was observed in the *irf8* mutant (control: 37% of examined animals showed bridging; YVAD: 68%). This rescue effect was comparable to that observed when Il-1β was suppressed for the entire 48 h (cf. Fig. 7d). LPS had no effect in wildtype or mutants during late regeneration. Hence, early inflammation and *il-1β* upregulation promote regeneration, but *il-1β* must be down-regulated at later phases of axonal regeneration.

**Fig. 6** *Tnf-α* is expressed by macrophages and regulates the immune response. **a** Top row: *tnf-α*:GFP labelling occurs almost exclusively in L-plastin[+] immune cells (L-plastin in green; *tnf-α*:GFP in magenta; yellow arrow indicates a rare *tnf- α*:GFP[+] microglial cell; 12 hpl) **b** In the injury site, the number and proportion of macrophages (Mfap4[+]) that are *tnf-α*:GFP[+] are much higher than numbers and proportions of microglia (4C4[+]) and neutrophils (Mpx[+]), indicating that the main source of Tnf-α is the macrophages. Arrows indicate double-labelled cells and arrowheads indicate immune cells that are *tnf-α*: GFP[-]. Single optical sections are shown; the proportion of macrophages that are *tnf-α*:GFP[+] decreases over time, whereas the proportion of *tnf-α*:GFP[+] microglial cells slightly increases (One-way ANOVA followed by Bonferroni post-*test*: $F_{4,195} = 376.3$, **$P < 0.01$, *$P < 0.05$). **c** Quantification of the immune cells after *tnf-α* gRNA injection shows that Tnf-α disruption leads to increased numbers of neutrophils (Mpx[+]) at 1 dpl but not at 2 dpl, whereas the numbers of macrophages/microglia (*mpeg1*:GFP[+]) remains unchanged (Mann–Whitney *U*-test: *$P < 0.05$, ns indicates no significance). **d** qRT-PCR indicates that *tnf-α* disruption leads to increased levels of *il-1β* mRNA at 2 dpl (*t*-tests: *$P < 0.05$). Lateral views of the injury site are shown; rostral is left. Scale bars: 50 μm. Error bars indicate SEM

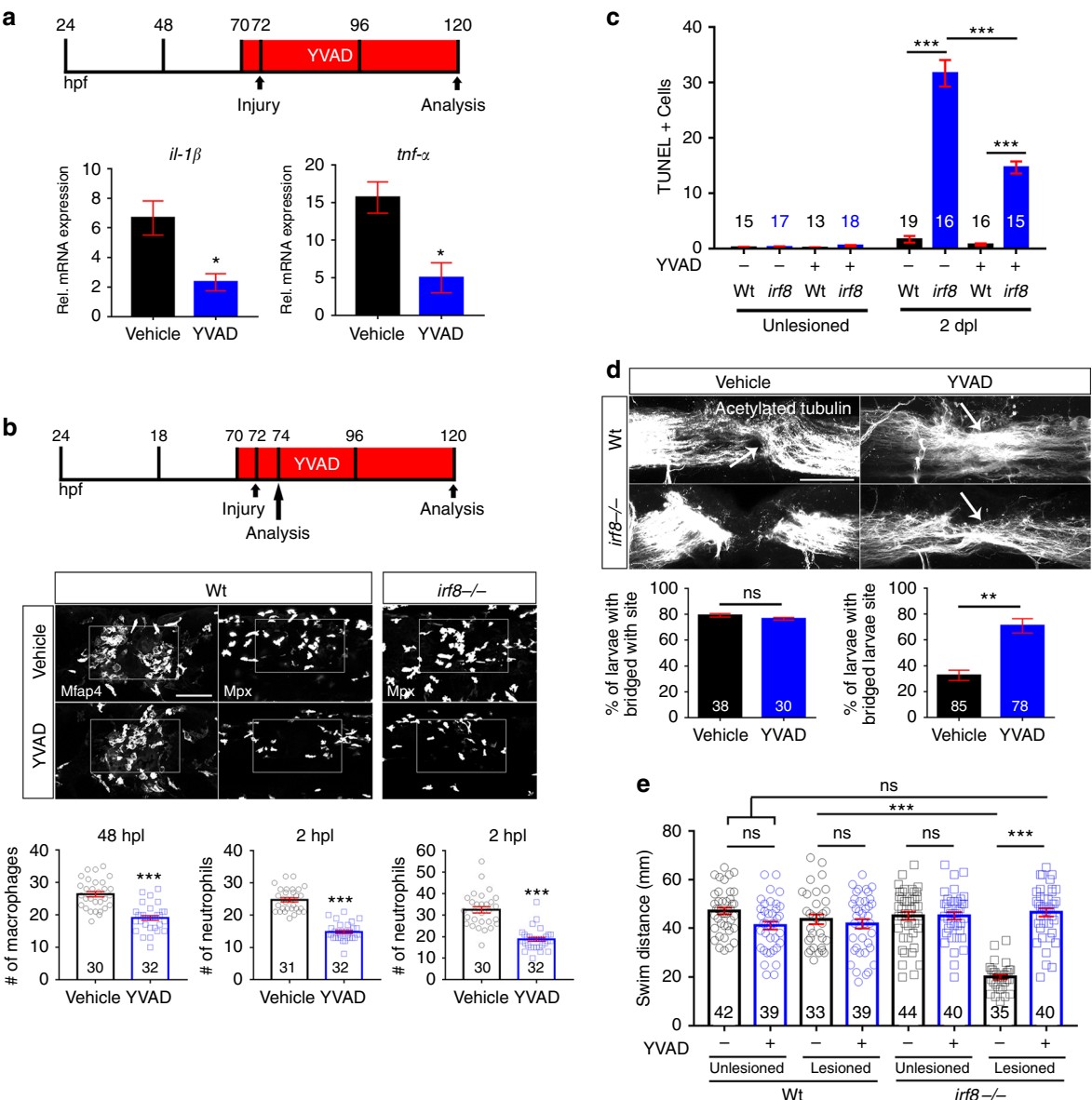

**Fig. 7** Inhibition of Il-1β function rescues axonal regeneration in *irf8* mutants. Lateral views of the injury site are shown; rostral is left. **a** YVAD reduces expression levels of *il-1β* and *tnf-α* in *irf8* mutants (two-sample *t*-test: *$P < 0.05$) at 2 dpl. **b** YVAD impairs migration of peripheral macrophages (Mfap4[+]) and neutrophils (Mpx[+]) in wildtype animals and *irf8* mutants (only neutrophils quantified, due to absence of macrophages) (*t*-tests: ***$P < 0.001$). **c** YVAD moderately reduces the number of TUNEL[+] cells in the *irf8* mutants at 2 dpl. (Two-Way ANOVA followed by Bonferroni multiple comparisons: $F_{3,121} = 112.5$, ***$P < 0.001$). **d** YVAD does not influence axonal regeneration in wildtype animals but rescues axonal bridging (arrows) in *irf8* mutants (Fisher's exact test: **$P < 0.01$, ns indicates no significance) at 2 dpl. **e** Impaired touch-evoked swimming distance in *irf8* mutants is rescued by YVAD treatment, to levels that are no longer different from lesioned and unlesioned wildtype animals at 2 dpl. YVAD has no influence on swimming distance in lesioned or unlesioned wildtype animals (Two-way ANOVA followed by Bonferroni multiple comparisons: $F_{1,309} = 35.229$, ***$P < 0.0001$, ns indicates no significance). Rectangle in **b** denotes quantification area. Scale bar: 50 μm for **b**, **d**. Error bars indicate SEM

**Reduction of Il-1β levels rescues swimming in *irf8* mutants**. To determine whether ll-1β inhibition also rescued recovery of swimming function in *irf8* mutants, we analysed touch-evoked swimming distances. YVAD had no effect in unlesioned mutant or wildtype animals and did not affect recovery in lesioned wildtype animals (Fig. 7e). In contrast, YVAD-treatment rescued the touch-evoked swimming distance in *irf8* mutants to levels that were indistinguishable from wildtype lesioned or unlesioned animals (Fig. 7e). Mean velocity and path shape (meandering) were also rescued (Supplementary Fig. 10I, J). These observations indicate that inhibition of Il-1β alone restores most axonal regeneration and recovery of touch-evoked swimming parameters in the absence of macrophages.

**Neutrophils are a major source of il-1β**. To understand *il-1β* regulation, we determined the source of Il-1β in wildtype and *irf8* mutants. Using Il-1β immunohistochemistry and a transgenic reporter line (*il-1β*:GFP), we found expression in microglia, macrophages, neutrophils and basal keratinocytes in the injury site (Fig. 8c, d; Supplementary 12A). Neuronal labelling (HuC/D[+]) did not overlap with *il-1β*:GFP labelling (Supplementary Fig. 12B). While numbers of *il-1β*:GFP[+] immune cells did not change significantly between 1 and 2 dpl, the percentage of macrophages (from 50.6 to 34.3%) and neutrophils that were labelled for the *il-1β*:GFP transgene were reduced (from 53.3 to

23.7%), likely reflecting resolution of inflammation (Supplementary Fig. 12A).

In *irf8* mutants, detection of *il-1β* mRNA by qRT-PCR and in situ hybridisation confirmed increased levels at 2 dpl, but not 1 dpl (Fig. 8a, b), despite lack of *il-1β* expressing microglia and macrophages. Instead, we observed increased numbers of Il-1β[+] neutrophils (by 97%) and basal keratinocytes (by 58%) compared to wildtype animals at 1 dpl (Fig. 8c, d). Importantly, the proportion of neutrophils that were Il-1β[+] was also increased from 33.6% in wildtype animals to 49% in the *irf8* mutant at 1 dpl. This demonstrates that neutrophils are more likely to express *il-1β* in the absence of macrophages.

Increased numbers of Il-1β[+] neutrophils in *irf8* mutants could, at least in part, be due to higher overall numbers of neutrophils in the injury site. We found a peak of neutrophil numbers in the lesion site of *irf8* mutants at 2 hpl, as in wildtype animals (Fig. 9a). However, the number of neutrophils was 27% higher than in wildtype, potentially due to the higher abundance of this cell type in the *irf8* mutant[18]. While neutrophil numbers declined over time in wildtype and *irf8* mutants, they did so more slowly in *irf8* mutants. At 24 hpl, twice, and at 48 hpl, three times the number of neutrophils as in wildtype animals remained in the mutant. Hence, macrophages control number of and cytokine expression by neutrophils[32], leading to prolonged presence of Il-1β[+] neutrophils in the injury site of *irf8* mutants.

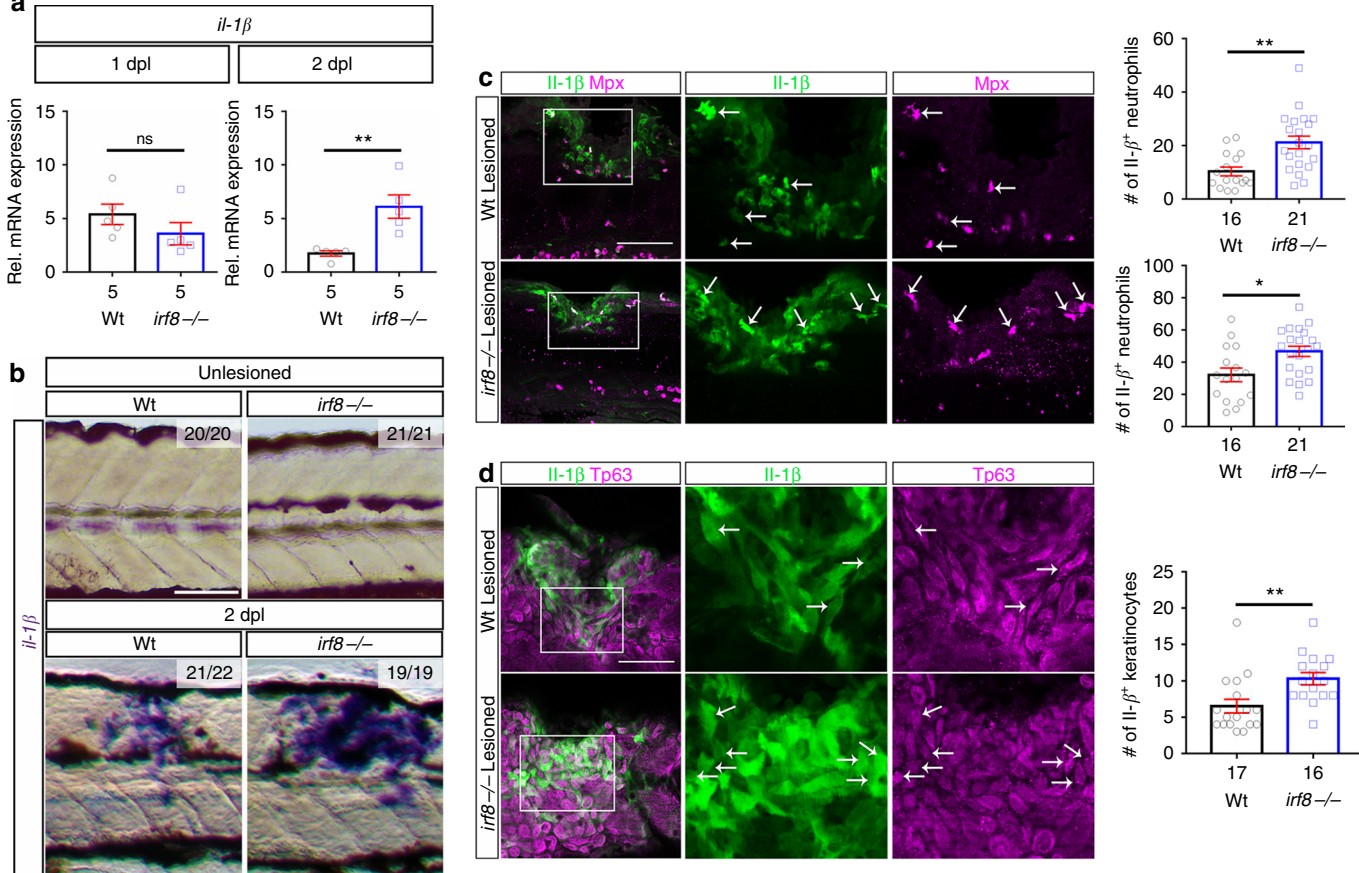

**Fig. 8** Levels of *il-1β* expression are increased in the injury site of *irf8* mutants. **a** At 1 dpl, expression levels of *il-1β* are comparable between *irf8* mutants and wildtype (Wt) animals but are higher in the mutant at 2 dpl in qRT-PCR (*t*-test: **$P < 0.01$, ns indicates no significance). **b** In situ hybridisation confirms increased expression of *il-1β* mRNA at 2 dpl. **c** In the injury site, the number and proportion of neutrophils (Mpx[+]) that are Il-1β immuno-positive (arrows) are increased in *irf8* mutants at 1 dpl compared to wildtype animals. **d** The number of basal keratinocytes (Tp63[+]) that are Il-1β immuno-positive is increased in *irf8* mutants. Single optical sections are shown; boxed areas are shown in higher magnifications (*t*-test: *$P < 0.05$, **$P < 0.01$). Lateral views of the injury site are shown; rostral is left. Scale bars: 100 μm in **b**, **c**, **d** and 50 μm for higher magnification areas. Error bars indicate SEM

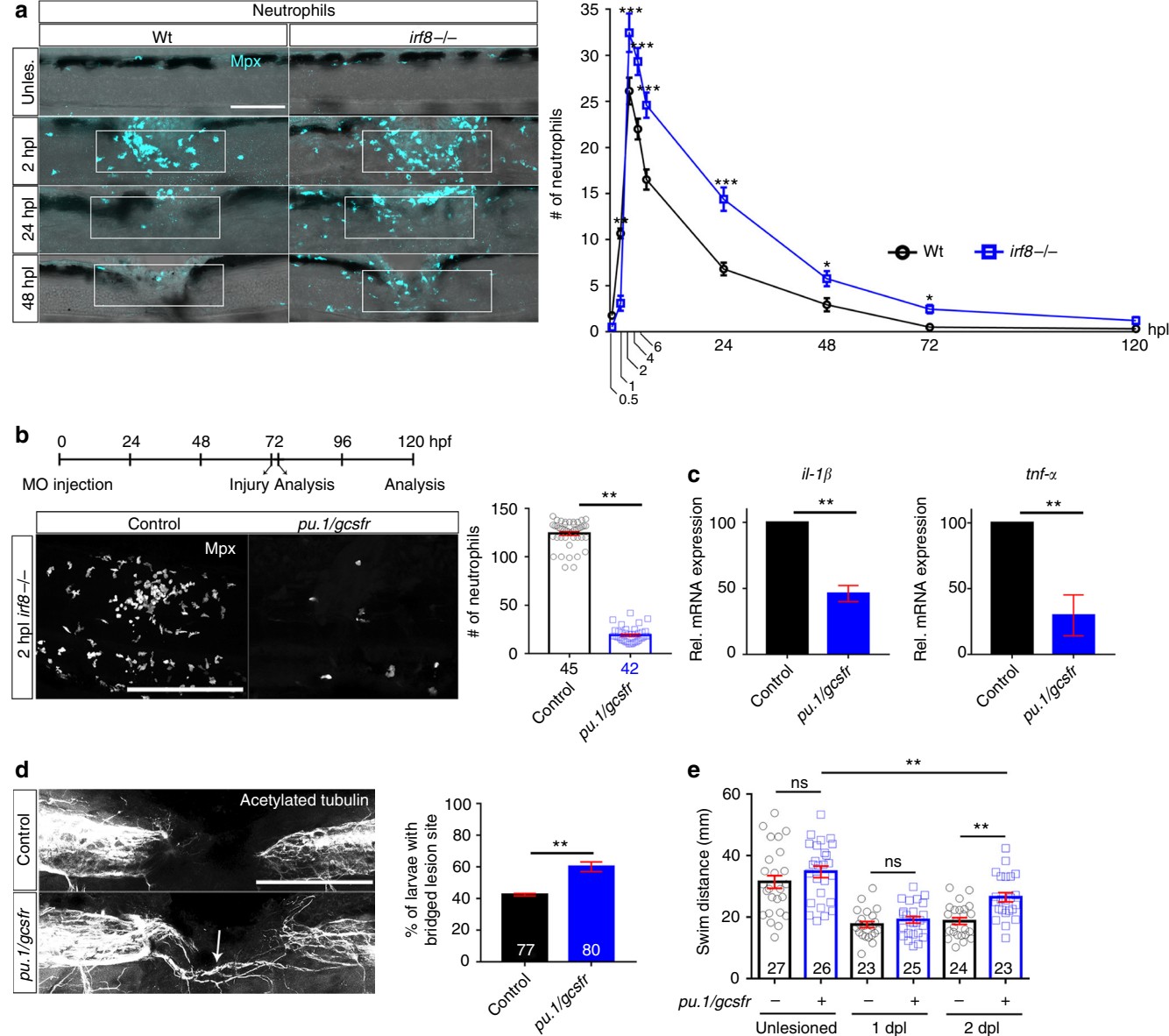

**Fig. 9** Preventing neutrophil formation partially rescues functional spinal cord regeneration in the *irf8* mutant. **a** In *irf8* mutants, higher peak numbers of neutrophils (Mpx[+]) at 2 hpl and slower clearance over the course of regeneration are observed (Two-Way ANOVA followed by Bonferroni multiple comparisons: $F_{8,427} = 13.19$ *$P < 0.05$, **$P < 0.01$, ***$P < 0.001$). Note that wildtype data are the same as shown in Fig. 1a, as counts in *irf8* mutants and wildtype animals were done in the same experiments. **b** Combination treatment with *pu.1* and *gcsfr* morpholinos efficiently prevents neutrophil accumulation in the lesion site (Mann–Whitney *U*-test: ***$P < 0.001$). **c** In *pu.1/gcsfr* morpholino injected *irf8* mutant fish, levels of *il-1β* and *tnf-α* mRNA expression are reduced at 2 dpl, as shown by qRT-PCR (*t*-test: ***$P < 0.001$). **d, e** In *pu.1/gcsfr* morpholino injected *irf8* mutant fish, axonal bridging (arrows, **d** Fisher's exact test: **$P < 0.01$) and behavioural recovery (**e** One-Way ANOVA followed by Bonferroni multiple comparisons: $F_{5,142} = 23.21$, **$P < 0.01$, ns indicates no significance) are partially rescued. Lateral views of the injury site are shown; rostral is left. Scale bars: 100 μm. Error bars indicate SEM

**Neutrophils inhibit regeneration in irf8 mutants**. To determine the relative importance of the neutrophils for regenerative failure in *irf8* mutants, we reduced their numbers using *pu.1/gcsfr* morpholino treatment. This strongly reduced numbers of neutrophils by 84.6% in the injury site at 2 hpl, when the neutrophil reaction peaked in untreated *irf8* mutants (Fig. 9b). *pu.1/gcsfr* morpholino treatment also reduced *il-1β* mRNA levels by 54% and *tnf-α* mRNA levels by 70% at 2 dpl (Fig. 9c). Remarkably, axon bridging (control: 43% of examined animals showed bridging; *pu.1/gcsfr* morpholino: 60%; Fig. 9d) and recovery of touch-evoked swimming distance were partially rescued in these neutrophil-depleted mutants at 2 dpl (Fig. 9e). This shows that in

the absence of macrophages, the prolonged presence of Il-1β[+] neutrophils is detrimental to regeneration.

## Discussion

We identify a biphasic role of the innate immune response for axonal bridging of the non-neural lesion site in larval zebrafish. Initial inflammation and Il-1β presence promote axon bridging, whereas later, Il-1β levels need to be tightly controlled by peripheral macrophages. Inhibiting Il-1β largely compensated for the absence of macrophages, underscoring the central role of this cytokine. Macrophage-derived Tnf-α promotes regeneration,

partially by reducing neutrophil number and il-1β levels (summarised in Supplementary Fig. 13). This indicates important and highly dynamic functions of the immune system for successful spinal cord regeneration.

The function of the immune system changes dramatically over time. Within the first hours after injury, neutrophils and the pro-inflammatory cytokines il-1β and tnf-α dominate the injury site. Initially, inflammation promotes axonal growth. This is indicated by reducing effects on axon bridging of early Il-1β inhibition and the promoting effect of LPS in wildtype animals. Indeed, it has been reported that Il-1β can promote neurite growth[33,34]. However, from about 12 hpl, macrophages and anti-inflammatory cytokines are in the lesion site and during this late phase of regeneration, Il-1β mediated inflammation in macrophage-less mutants strongly inhibits regeneration.

How does Il-1β inhibit spinal cord regeneration? High Il-1β levels may condition the environment to be inhibitory to axonal regrowth. Blocking excessive Il-1β signalling in irf8 mutants revealed that Il-1β increases numbers of neutrophils, levels of il-1β and tnf-α expression, as well as cell death in the lesion environment. Moreover, reduced expression of some metalloproteinases in the irf8 mutant suggest that the lesion site ECM may be altered. However, several ECM components, including functionally important Col XII[1], were unaltered in expression in the irf8 mutant. Similar to our observations, an Il-1β deficient mouse showed slightly increased axonal regrowth after spinal injury[35].

Axonal regeneration is promoted by Tnf-α. Even though il-1β and tnf-α were similarly upregulated in the irf8 mutant, only reducing Il-1β levels rescued the mutant. Conversely, in wildtype animals, Il-1β had only a relatively small promoting effect on early regeneration, whereas Tnf-α was indispensable for axonal regrowth. Different functions for the two pro-inflammatory cytokines have been reported[36]. They also differ in cell type of origin. Whereas Il-1β is expressed by a substantial proportion of neutrophils, microglia and macrophages, Tnf-α is mainly expressed by macrophages in the injury site. This indicates a clear difference between tissue-resident microglia and peripheral macrophages. In mammals, tnf-α is produced by both microglia and macrophages[37].

Tnf-α may exert its positive role for regeneration at least in part by controlling neutrophil numbers and Il-1β levels. Both of these parameters are increased when Tnf-α is inhibited and are inhibitory to regeneration. Anti-inflammatory actions of Tnf-α have been described in the context of auto-immunity[38], but whether this interaction between Tnf-α and Il-1β is direct or indirect, needs to be elucidated. For example, Tnf-α can be neuroprotective after CNS injury[36,39] and thus indirectly reduce inflammation. In the regenerating fin, Tnf-α has an important promoting function for blastema formation[40]. This suggests that Tnf-α may be involved in remodelling repair cells in the lesion site after spinal injury, which then creates an axon growth-promoting environment.

The role of Tnf-α for axonal regeneration in mammals is not clear. Some reports indicate axon growth promoting properties of Tnf-α[41,42], whereas others show inhibition of axon growth[43]. Negative effects of Tnf-α on lesion-induced cell death[44] and functional recovery[37,45] have also been reported. However, knock out of Tnf-α had no reported effect after spinal injury[46].

Preventing neutrophil formation in irf8 mutants, indicates that il-1β expressing neutrophils are major mediators of the inhibitory immune response in the absence of microglia and macrophages. However, the rescue of axon regrowth and swimming function was only partial. This could be explained by the absence of the early regeneration-promoting influence of the inflammation or basal keratinocytes still expressing il-1β in neutrophil-depleted

irf8 mutants. In mammals, neutrophils cause secondary cell death[47,48] and depleting neutrophils leads to favourable injury outcomes[5], similar to our observations.

Macrophages control inflammation, as their absence in irf8 mutants leads to abnormally high expression levels of pro-inflammatory cytokines il-1β and tnf-α. This is similar to observations in fin regeneration[49]. In the absence of macrophages, positive feedback regulation of il-1β takes place, as indicated by more il-1β positive neutrophils and basal keratinocytes in the irf8 mutants and reduced il-1β mRNA levels when Il-1β function was inhibited. Moreover, a higher proportion of neutrophils were Il-1β+ in irf8 mutants, showing that without macrophages, neutrophils have a more pro-inflammatory phenotype. We show that macrophages/microglia, together with other tissues, express anti-inflammatory cytokines tgf-β1a and tgf-β3 and could thus be partly responsible for reducing pro-inflammatory phenotypes in wildtype animals.

Macrophages do not promote regeneration primarily by preventing cell death or removing debris. We observed phagocytosing macrophages by time-lapse imaging and debris levels were clearly increased it the absence of macrophages in irf8 mutants. However, when axon regrowth was rescued in the mutant by Il-1β inhibition, debris levels were still higher than in controls. Preventing cell death did not rescue axon growth and inhibition phagocytosis in wildtype animals did not impair regeneration. Hence, regenerative success does not correlate with debris abundance. Interestingly, in fin regeneration, lack of macrophages also leads to increased cell death. As this leads to death of tissue progenitor cells, fin regeneration is inhibited[49]. In mammalian spinal injury, debris, especially myelin debris, is inhibitory to regeneration[50].

Are macrophages the most important immune cell type for axonal regrowth? Unimpaired axonal regrowth in the csfr1a/b mutant, in which microglial cells are absent and neutrophils are strongly reduced in number, indicates that these cells may be dispensable for regeneration. However, the increase in peripheral macrophages in this mutant could have compensated for a possible regeneration-promoting role of microglia. Since some neutrophils are still present in the injury site in csfr1a/b mutants, these might contribute to promoting axonal regrowth.

Endothelial cells and myelinating cells are unlikely to be major mediators of early regeneration in larval spinal cord regeneration. Endothelial cells from injured blood vessels were slow to reform blood vessels and were rarely invading the lesion site. In contrast, in mammals endothelial cells accumulate in the injury site, where they may have anti-inflammatory functions[51]. Myelinating cells bridged the lesion site, but were not abundant and only did so, when axons had already crossed the lesion site. Although relatively late, axons become remyelinated, which may contribute to recovery of some swimming parameters after injury. In mammals, transplanted myelinating cells, such as olfactory ensheathing cells and Schwann cells have been shown to improve recovery after spinal injury[52,53].

Astroglia-like processes cross the injury site and this depends on the immune response, as we show here. While these processes cross the injury site independently of and slightly later than axons and axons still cross when these cells are ablated[1], astroglia-like processes produce growth factors that support axonal regeneration[54,55].

Timing of the immune response is crucial for regenerative success after spinal lesion. Macrophages in mammals[3,37] and zebrafish[56] display pro-inflammatory and anti-inflammatory phenotypes and the anti-inflammatory phenotypes are seen as beneficial for regeneration[7–9]. We show that inflammation is rapidly downregulated in zebrafish concurrent with the upregulation of anti-inflammatory cytokines, which does not readily occur in mammals[3].

In summary, we have established an accessible in vivo system to study complex interactions of immune cells and a spinal injury site in successful regeneration. This allows fundamental insight into the role of immune cells that may ultimately inform non-regenerating systems. Here, we demonstrate a pivotal role of macrophages in promoting functional spinal cord regeneration, by producing Tnf-α and controlling Il-1β-mediated inflammation.

## Methods

**Animals**. All zebrafish lines were kept and raised under standard conditions[57] and all experiments were approved by the British Home Office (project license no.: 70/8805). Regeneration proceeds within 48 h of the lesion, therefore most analyses of axonal regrowth, cellular repair, and behavioural recovery can be performed before the fish are protected under the A(SP)A 1986, reducing the number of animals used in regeneration studies following the principles of the 3 rs. Approximately 11,000 larvae of either sex were used for this study, of which 8% were over 5 dpf.

The following lines were used: WIK wild type zebrafish, Tg(*Xla.Tubb*:DsRed)[zf14826], abbreviated as *Xla.Tubb*:DsRed[58]; Tg(*mpeg1*:EGFP)[gl22], abbreviated as *mpeg1*:GFP[59], and Tg(*mpx*:GFP)[uwm1], abbreviated as *mpx*:GFP[60], Tg(*fli1*:EGFP)[y1], abbreviated as *fli1*:GFP[61]; *irf8*[st95/st95], abbreviated as *irf8* mutants[18]; *csf1ra*[j4e1/j4e1] × csf1rb[+/re01] incrosses, phenotypically sorted for absence of 4C4+ cells in the head, abbreviated as *csfr1a/b* mutants[19]; TgBAC(*pdgfrb*:Gal4FF)[ncv24]; Tg(*UAS*:GFP), abbreviated as *pdgfrb*:GFP[62],Tg(*6xTCF/LefminiP*:2dGFP), abbreviated as *6xTCF*:dGFP[63], Tg(*claudin k*:Gal4)[ue101]; Tg(14x*UAS*:GFP) abbreviated as *cldnK*:GFP[64], Tg(*tnfa*:eGFP-F)[sa43296], abbreviated as *tnf-α*:GFP[56] and Tg(*il-1β*:eGFP)[sh445], abbreviated as *il-1β*:GFP[65].

**Drug treatment**. Dexamethasone (Dex) (Sigma, Gillingham, UK) was dissolved in DMSO to a stock concentration of 5 mM. The working concentration was 10 μM prepared by dilution from stock solution in fish water. Ac-YVAD-cmk (YVAD) (Sigma) was dissolved in DMSO to a stock concentration of 10 mM. The working concentration was 50 μM prepared by dilution from the stock solution in fish water. Q-VD-OPh (Sigma), abbreviated as QVD in the manuscript, was dissolved in DMSO to a stock concentration of 10 mM. The working concentration was 50 μM. O-Phospho-L-serine (L-SOP) (Sigma) was dissolved in PBS to a stock concentration of 10 mM. The working concentration was 10 μM prepared by dilution from stock solution. Lipopolysaccharides from Escherichia coli O55:B5 (LPS, Sigma) were dissolved in PBS to a stock concentration of 1 mg/ml. The working dilution was 50 μg/ml. Pomalidomide (Cayman Chemicals, Michigan, USA) was diluted in DMSO at a stock concentration of 10 mg/ml. For the treatments, 6.9 μl of the stock where diluted in 1.5 ml of fish water. Larvae were pre-treated for 2 h before the injury and were incubated for 24 and 48 hpl. Larvae were collected from the breeding tanks and were randomly divided into Petri dishes at a density of maximally 30 larvae per dish, but no formal randomisation method was used. For most drug treatments, larvae were incubated with the drug from 3 dpl until 5 dpl, if not indicated differently.

**Spinal cord lesions**. Zebrafish larvae at 3 dpf were anaesthetised in PBS containing 0.02% aminobenzoic-acid-ethyl methyl-ester (MS222, Sigma), as described[1]. Larvae were transferred to an agarose-coated petri dish. Following removal of excess water, the larvae were placed in a lateral position, and the tip of a sharp 30 ½ G syringe needle was used to inflict a stab injury or a dorsal incision on the dorsal part of the trunk at the level of the 15th myotome.

**Behavioural analysis**. Behavioural analysis was performed as previously described[14]. Briefly, lesioned and unlesioned larvae were touched caudal to the lesion site using a glass capillary. The swim distance of their escape response, the mean velocity and the meandering were recorded for 15 s after touch and analyzed using a Noldus behaviour analysis setup (EthoVision version 7). Data given is averaged from triplicate measures per fish. Between repeated measures, the larvae were left to recover for 1 min. The observer was blinded to the treatment during the behavioural assay. The assay was performed on five independent clutches in order to assess the behavioural recovery in the *irf8* mutants and three independent clutches after the YVAD treatment.

**Fluorescence-activated cell sorting**. Macrophages and microglia were isolated from 4 dpf transgenic *mpeg1*:eGFP embryos by FACS. For this purpose, about 500 fish were lesioned by transecting the spinal cord. Trunk-containing lesion site were dissected and collected at 24 hpl and used for cell dissociation[66]. Cells purified after FACS were used for qRT-PCR.

**Quantitative RT-PCR**. RNA was isolated from the injury sites of the larvae using the RNeasy Mini Kit (Qiagen, Hilden, Germany). Forty larvae were used for each condition. cDNA used as template was created using the iScript™ cDNA Synthesis Kit (Bio-Rad, Munich, Germany). Standard RT-PCR was performed using SYBR Green Master Mix (Bio-Rad). qRT-PCR was performed at 58 °C using Roche Light Cycler 96 (Roche Diagnostics, West Sussex, UK) and relative mRNA levels

determined using the Roche Light Cycler 96 SW1 software. Samples were run in duplicates and expression levels were normalised to β-actin control. Primers were designed to span an exon-exon junction using Primer-BLAST. Sequences are given in supplementary Table 2. All experiments were carried out at least as biological triplicates.

**In situ hybridisation**. For whole mount in situ hybridisation[1], after fixation in 4% PFA, larvae were digested with 40 μg/ml Proteinase K (Invitrogen, Carlsbad, USA). Thereafter, larvae were washed briefly in PBT and were re-fixed for 20 min in 4% PFA followed by washes in PBT. After washes, larvae were incubated at 67 °C for 2 h in pre-warmed hybridisation buffer. Hybridisation buffer was replaced with digoxigenin (DIG) labelled ISH probes diluted in hybridisation buffer and incubated at 67 °C overnight. The next day, larvae were washed thoroughly at 67 °C with hybridisation buffer, 50% 2× SSCT/50% deionized formamide, 2x SSCT and 0.2x SSCT. Larvae were then washed in PBT and incubated for 1 h in blocking buffer under slow agitation. Thereafter, larvae were incubated overnight at 4 °C in blocking buffer containing pre-absorbed anti-DIG antibody. The next day, larvae were washed in PBT, followed by washes in staining buffer. Colour reaction was performed by incubating larvae in staining buffer supplemented with NBT/BCIP (Sigma-Aldrich) substrate. The staining reaction was terminated by washing larvae in PBT.

**Immunofluorescence**. All incubations were performed at room temperature unless stated otherwise. Antibodies used are listed in supplementary Table 3. For most immunolabelling experiments, the larvae were fixed in 4% PFA-PBS containing 1% DMSO at 4 °C overnight. After washes in PBS, larvae were washed in PBTx. After permeabilization by incubation in PBS containing 2 mg/ml Collagenase (Sigma) for 25 min larvae were washed in PBTx. They were then incubated in blocking buffer for 2 h and incubated with primary antibody (1:50–1:500) diluted in blocking buffer at 4 °C overnight. On the following day, larvae were washed times in PBTx, followed by incubation with secondary antibody diluted in blocking buffer (1:300) at 4 °C overnight. The next day, larvae were washed three times in PBTx and once in PBS for 15 min each, before mounting in glycerol.

For whole mount immunostaining using primary antibodies from the same host species (rabbit anti-Il-1β, rabbit anti-Mpx, rabbit anti-Tp63) the samples were initially incubated with the first primary antibody at 4 °C overnight. After washes with PBTx the samples were incubated with the conjugated first secondary antibody overnight at 4 °C. Subsequently, samples were incubated with blocking buffer for 1 h at RT in order to saturate open binding sites of the first primary antibody. Next, the samples were incubated with unconjugated Fab antibody against the host species of the primary antibody in order to cover the IgG sites of the first primary antibody, so that the second secondary antibody will not bind to it. After this, samples were incubated with the second primary antibody overnight at 4 °C and subsequently with the second conjugated secondary antibody overnight at 4 °C before mounting in glycerol. No signal was detected when the second primary antibody was omitted, indicating specificity of the consecutive immunolabeling protocol.

For whole mount immunostaining of acetylated tubulin[1], larvae were fixed in 4% PFA for 1 h and then were dehydrated and transferred to 100% MeOH and then stored at −20 °C overnight. The next day, head and tail were removed, and the samples were incubated in pre-chilled Acetone. Thereafter, larvae were washed and digested with Proteinase K and re-fixed in 4% PFA. After washes the larvae were incubated with BSA in PBTx for 1 h. Subsequently the larvae were incubated for 2 overnights with primary antibody (acetylated tubulin). After washes and incubation with the secondary antibody the samples were washed in PBS for 15 min each, before mounting in glycerol.

**Evaluation of cell death using acridine orange**. In order to assess the levels of cell death after injury we used the acridine orange live staining as described by others[67]. Briefly, at 1 and 2 dpl the larvae were incubated in 2.5 μg/ml solution of dye diluted into conditioned water for 20 min. After the staining, the larvae were washed by changing the water and larvae were live-mounted for imaging.

**Identification of dying cells after injury**. In order to assess the levels of cell death after injury cross sections of larvae were used. Larvae were fixed in 4% PFA overnight at 4 °C. After washes with 0.5% PBSTx, the larvae were transferred to 100% methanol and incubated for 10 min at room temperature. After rehydration, the larvae were washed with PBSTx 0.5%. Following this, larvae were mounted in 4% agarose and 50 μm sections were performed using a vibratome (MICROM HM 650 V, VWR, Leicestershire, UK). The sections were then permeabilized using 14 μg/ml diluted in 0.5% PBSTx. After brief wash with PBSTx the sections were postfixed in 4% PFA for 20 min. Excess PFA was washed out and the samples were incubated with the TUNEL reaction mix according to the In-situ Cell Death Detection Kit TMR red protocol (Roche).

**Western blotting**. Zebrafish larvae were sacrificed by an overdose of MS-222 at 4 dpf and used for protein extraction. Around 60 fish per condition were homogenised in 250 μl of 1x PBS/1% Triton X-100 (containing protease inhibitor cocktail complete, Roche Diagnostics), using a tissue grinder. After 1 h incubation

at 4 degrees, lysates were centrifuged at 12,000 rpm for 20 mins to remove Triton-X-100-insoluble debris. Protein concentrations were quantified using a BCA assay following manufacturer's instruction and the same amount of protein for each sample was loaded on a denaturing 12% acrylamide gel. After the electrophoretic run, the proteins were transferred on a nitrocellulose membrane (BioRad, Germany) and, after 1 h blocking in PBS/5% non-fat dry milk/0.1% Tween-20, probed with either rabbit anti-TNFα (1:2000, Anaspec; Fremont, CA), rabbit anti IL-1 (1:200, Proteintech, Manchester, UK) or mouse anti-α Tubulin (1:2000, DSHB) O/N at 4 degrees. Goat anti-rabbit (IRdye680LT, LI-COR, Lincoln, Nebraska, USA) and goat anti-mouse (IRdye800CW, LI-COR) secondary antibodies were used and the signal was detected by an Odyssey (LI-COR) imaging system and analysed with the Image Studio Lite software version 5.2.

**Morpholino injection**. All morpholinos were injected into single cell stage larvae in total volume of 2 nl. Knockdown of *il-1β* was carried out using the antisense morpholino against *il-1β* (5′-CCCACAAACTGCAAAATATCAGCTT-3′), targeting the splice site between intron 2 and exon 3 according to[31]. In order to block neutrophil development, we used the previously described MO combination of *pu.1* (5′-GATATACTGATACTCCATTGGTGGT-3′) which targets the translational start (ATG) of the pu.1 and the splice blocking MO against *gcsfr* (5′-TTTGTCTTTACAGATCCGCCAGTTC-3′)[16]. All morpholinos and standard control (5′-CCTCTTACCTCAGTTACAATTTATA-3′) were obtained from Gene Tools, LLC, Oregon, USA.

**CRISPR-mediated genome editing**. CRISPR gRNA for *il-1β* was designed using CRISPR Design (http://crispr.mit.edu) and ZiFit (http://zifit.partners.org/ZiFiT) webtools. Vectors were generated by ligating the annealed oligonucleotides into the pT7-gRNA expression vector[1]. The gRNA was transcribed using the mMESSAGE mMACHINE T7 kit (Ambion, ThermoFisher SCIENTIFIC, Loughborough, UK) and assessed for size and quality on an electrophoresis gel. The injection mix consisted of 75 pg *il-1β* gRNA (target sequence: 5′-TGTGGAGCGGAGCCT TCGGCGGG-3′) and 150 pg Cas9 RNA and was injected into single cell stage larvae. The CrRNA for *tnf-α* (target sequence: 5′-CCCGATGATGGCATTTA TTTTGT-3′) and the tracrRNA were ordered from Merck KGaA (Germany, Darmstadt). The injection mix included 1 μl Tracer 250 ng/μl, 1 μl gRNA, 1 μl Cas9 protein, 1 μl RNAse free H$_2$0, 1 μl fluorescent dextran. Larvae injected with GFP gRNA (target sequence: 5′GGCGAGGGCGATGCCACCTA-3) and uninjected larvae were used as controls. The efficiency of the mutagenesis was assessed by RFLP analysis.

**Live imaging of zebrafish larvae and time-lapse imaging**. For the acquisition of all fluorescent images, LSM 710 and LSM 880 confocal microscopes were used. For live confocal imaging, zebrafish larvae were anesthetized in PBS containing 0.02% MS222 and mounted in 1.5% low melting point agarose (Ultra-PureTM, Invitrogen). During imaging, the larvae were covered with 0.01% MS222-containing fish water to keep preparations from drying out. For time-lapse imaging, agarose covering the lesion site was gently removed after gelation. Time-lapse imaging was performed for 19 h starting at 6 hpl. Acquired time-lapse images were denoised using the ImageJ plugin CANDLE-J algorithm. Comparison of raw movies with CANDLE-J-processed movies showed that edges of features remained conserved after denoising.

**Scoring and measurement of spinal cord bridging**. Axonal and astroglial bridging was scored in fixed and live mounted samples using fluorescence-equipped stereomicroscope (Leica MDG41) and confocal microscopes (LSM 710, LSM 880) at time points of interest as previously described[1]. Any continuity of labelling between the two spinal cord stumps was scored as "bridged". In some cases, the thickness of the axonal bridge was measured in collapsed confocal image stacks, by determining the length of (a) vertical line(s) that covers the width of crossing fascicles at the centre of the injury site. The observer was blinded to the experimental condition before scoring or measuring and experiments were performed blinded to the experimental condition on at least three independent clutches of larvae.

**Cell counting in whole-mounted larvae**. A volume of interest was defined centred on the lesion site from confocal images. The dimensions were: width = 200 μm, height = 75 μm (above the notochord), depth = 50 μm. Images were analysed using the Imaris (Bitplane, Belfast, UK) or ImageJ software. The number of cells was quantified manually in 3D view, blinded to the experimental condition on at least three independent clutches of larvae.

**Quantification of diffuse signals in whole-mount larvae**. For most quantifications of diffuse signal, e.g., acridine orange, image stacks, centred in the lesion site (height: 50 μm) were collapsed and an area of interest defined: width = 100 μm, height = 50 μm (above the notochord). The image was thresholded and the "Analyze Particles" tool in Fiji with default settings was used to calculate the percentage of area taken up by signal.

To quantify Col I and Tnf-α in the injury site (Supplementary Figs 4C and 9A) we used a previously published protocol[1]. Briefly, image stacks were collapsed, thresholded and the area of the signal was measured.

All analyses were performed blinded to the experimental condition on at least three independent clutches of larvae.

**Statistical analysis**. Power analysis using G*Power[68], was used to calculate power (aim > 0.8) for the experiments and determine the group sizes accordingly. Statistical power was >0.8 for all experiments. All quantitative data were tested for normality and analyzed with parametric and non-parametric tests as appropriate. The statistical analysis was performed using IBM SPSS Statistics 23.0. Shapiro-Wilk's *W*-test was used in order to assess the normality of the data. Quantitative RT-PCR data were analyzed as previously described[69,70]. Kruskal–Wallis test followed by Dunn's multiple comparisons, One-way ANOVA followed by Bonferroni multiple comparisons test, two-way ANOVA, followed by Bonferroni multiple comparisons, *t*-test, Mann–Whitney *U*-test or Fischer's exact test were used, as indicated in the figure legends. *$P < 0.05$, **$P < 0.01$, ***$P < 0.001$, n.s. indicates no significance. Error bars indicate the standard error of the mean (SEM). The Figures were prepared with Adobe Photoshop CC and Adobe Illustrator CC. Graphs were generated using GraphPad Prism 7.

## Data availability
The authors declare that all data supporting the findings of this study are available within the article and its supplementary information files, or from the corresponding authors on reasonable request.

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

## Acknowledgements

We thank Dr. William Talbot and Dr. Stephen Renshaw for sharing mutants before publication, Dr. Bertrand Vernay for expert advice on microscopy and Joly Ghanawi for excellent fish facility management. Supported by the BBSRC to C.G.B. and T.B., NC3Rs PhD studentship to C.G.B./T.B. for T.M.T., DFG to D.W., Summer research studentships

to T.M. (Carnegie Trust) and EK (Anatomical Society). N.O. was funded by Biotechnology and Biological Sciences Research Council (BBSRC) project grant (BB/L000830/1). T.M.T. and M.K. are currently being paid by Biogen who did not have any influence on this study.

## Author contributions

Conceptualisation–T.M.T., T.Be. and C.G.B.; Investigation–T.M.T., D.W., L.C., T.M., M.K., M.L., A.U., T.Ba. and E.K.; Resources–N.O., S.A.R., Y.F. and T.J.H.; Supervision–D.W., N.O., Y.F., T.Be. and C.G.B.; Writing–T.M.T., T.Be. and C.G.B.; Funding acquisition–T.Be. and C.G.B.

## Additional information

**Competing interests:** The authors declare no competing interests.

