## [Peer Review File · Nature Communications]

Reviewers' Comments:

Reviewer #1:

Remarks to the Author:

The manuscript by Tsarouchas et al. reports the important role that macrophages play in spinal cord axonal regeneration by producing pro-regenerative $\text{tnf-}\alpha$ and controlling the levels of the pro-inflammatory cytokine $\text{Il-1}\beta$. This work is of general interest and it brings a better understanding on the role of immune cell responses and inflammation after spinal cord injury in a successfully regenerating species. The studies presented are based on in vivo imaging in zebrafish in combination with the use of genetic and pharmacologic tools allowing for the manipulation of subpopulations of innate immune cells. Briefly, Tsarouchas et al. show that after complete spinal cord transection, macrophages, but not neutrophils or microglia, are necessary for functional spinal cord regeneration. All of the experimental timelines are clearly presented and the statistical analyses are done rigorously. The conclusions are generally supported by the data described in the paper, however, the language used by the authors often overstates their results and should be more reflective of the findings they report in the paper. While the authors' work is generally rigorous and of great quality, there are points that could be improved regarding the use of CRISPR/cas9, and some changes are needed in the main text. I lay out some major concerns below:

1. All throughout the article, the authors quantify the "percentage of larvae with bridged lesion site", but do not examine the efficiency of the axonal bridging per se (e.g. quantification of the number of axons that fully regrew or measurement of the tubulin fluorescence intensity across the lesion site). Their method unfortunately does not take into account the variability in bridging among fish or among different experimental conditions and does not support their claim of "full regeneration" of the spinal cord. Adding a more rigorous assessment of axonal bridging would greatly improve the significance of the findings and may reveal more subtle changes in the regenerative capacity with regards to some of their manipulations.
2. Tsarouchas et al. claim that macrophages are necessary and sufficient for full regeneration. I think this conclusion is overstated. Their work shows axonal bridging across the lesion site up to 5 days post lesion, but it by no means shows full regeneration of the spinal cord. This study does not assess the presence of glial cells associated with axons post-bridging and does not show if lesioned axons eventually get remyelinated. The authors also do not present any satisfactory behavioral assay post-lesion, which should be demonstrated in order to claim a full regeneration of the spinal cord. Fig. 7E shows that inhibition of $\text{Il-1}\beta$ "restores functional recovery" and "rescues impaired touch-evoked swimming" in lesioned irf8-/- , however, the measurement of the swimming distance alone does not reflect the swimming pattern or speed. A more detailed assessment of swimming in these studies is important if claiming full functional recovery.
3. For the purpose of validating the pharmacological inhibition of $\text{TNF-}\alpha$, the authors generated somatic mutations in $\text{tnf-}\alpha$ by using the CRISPR/Cas9 genome editing tool and examined the percentage of injected larvae with a bridged lesion site after spinal cord transection (Fig. 6 B and C). It is unclear however, whether all of the $\text{TNF-}\alpha$ gRNA injected larvae were individually genotyped and tested for somatic mutations, as the authors do not score the efficiency of the Cas9 cleavage. Although the RFLP analysis is a validated way to show that Cas9 produces cuts in the gene of interest, the authors do not provide any clonal analysis of the somatic mutations generated in the F0 larvae and do not prove that the cuts lead to indels. CRISPR/Cas9 injected embryos should be characterized more thoroughly and this requires more controls.
4. In Fig. 6C, Tsarouchas et al. show that axonal bridging is impaired after disruption of $\text{tnf-}\alpha$. The authors do not rule out the possibility that $\text{TNF-}\alpha$ gRNA injected larvae may display a regenerative delay due to the injection of gRNA and cas9 mRNA, that often leads to developmental delays, and therefore should provide better controls. The authors only examined the injected larvae at 1dpl and 2dpl, while they could have examined the phenotype at 5dpl as they did with other mutants

earlier in the paper (e.g. irf8^{-/-} in Fig.2B).

5. The authors show that TNF- α :GFP is mostly expressed in macrophages (95.7%) in the lesion site, as supported by the immunostainings for L-plastin, 4C4 and Mpx presented in Fig.6D. However, the authors do not mention in the manuscript that another source of TNF- α could potentially be axons themselves. At least one previous study has demonstrated that sensory axons express tnfa. The intrinsic expression of TNF- α in sensory axons may participate in creating a favorable environment for regeneration. For this reason, the use of the adjective "sufficient" seems inadequate.

6. Studies in adult zebrafish demonstrate the essential role that radial glia play in bridging spinal cord lesion sites and creating a scaffold for axonal regrowth. In larvae that are not deprived of radial glia, can the authors say that "immune system is necessary for and sufficient to promote axon regeneration"? Here again, the vocabulary may not be appropriate as radial glia may also be involved in their studies.

7. By doing qPCR analysis, the authors show a decrease in expression of the anti-inflammatory cytokines TGF- β 1a and TGF- β 3 during the late regenerative phase in irf8^{-/-}. The authors propose that TGF- β come from macrophages, but Fig.5C shows that the level of TGF- β 1a peaks 15 minutes post-lesion in irf8^{-/-}. This raises the question of the source of TGF- β other than macrophages.

Reviewer #2:

Remarks to the Author:

The manuscript highlights the pivotal role of macrophages in axonal regeneration in the zebrafish. The work is elegant, and the results are, on the whole, convincing.

Several comments:

1. The manuscript is somewhat overloaded with information. Some of the results are negative, and do not directly support the overall conclusion. Such results should be placed in the supplementary section.
2. Regarding experiments to determine the effect of macrophages on ECM composition- macrophages are the most obvious source of ECM degrading enzymes, and therefore the authors should consider testing a battery of enzymes that can potentially degrade the ECM
3. The role of the phagocytic function of macrophages should be tested.
4. Many of the literature citations are inaccurate.
5. More insight is needed regarding the underlying role of TNFa.
6. A more comprehensive discussion should be devoted to the issue of how inflammatory associated cytokine have distinct and opposite role in repair.
7. The Scheme of the model is not self-explanatory; the temporal issue should be emphasized.

Reviewer #3:

Remarks to the Author:

This manuscript by Tsarouchas et al describes their results from a comprehensive analysis of the role of innate immune responses in axon regeneration in zebrafish. An initial finding is that macrophage-deficient mice (irf8 mutants) exhibit defects in axon regeneration, and this appears to be largely mediated by peripheral macrophages, instead of microglia. An important underlying regenerative mechanism is the prolonged inflammation after injury, with elevated levels of IL1 β and TNF α . Their further studies showed that these factors differ in their role in impacting axon regeneration, with TNF α shown to promote, and IL1 β shown to be binary (initially promoting and later inhibiting). Most conclusions were supported by more than one line of evidence and are largely convincing to me. I think that these results together provide an interesting and comprehensive picture of the involvements of innate immunity in the processes of

spinal cord injury and axon regeneration in the fish model. There are a few issues to be fixed prior to publication.

Some of manipulations need better additional verification. For example, several methods were used to block TNFalpha, but to what extents TNFalpha's expression was reduced was not assessed.

The authors indicated a distinct role of peripheral macrophages and microglia. The question is whether this is a quantitative or a qualitative difference. Are there any differences in terms of their gene expression relevant to this context?

The relationship between axon bridge and swimming behavioral recovery: From the text, axon bridge formation and swimming recovery appear with a similar temporal window. Assuming that regenerated axons need to make synapses, it is unclear whether behavioral recovery is entirely due to regenerating axons. What portions of axon regeneration are required for swimming behavioral? Because multiple manipulations affect axon regeneration to different extents in this manuscript, it will be useful for the authors to analyze whether there is a clear correlation.

Finally, this manuscript could be improved by additional editing. There are a number of errors in this current manuscript. For example, the legend for Fig. 1E and I are missing. This is an important missing point, because it is unclear how axonal bridges were assessed in this case and whether it is the same method as other figures.

RESPONSE TO REVIEWERS

Reviewer #1 (Remarks to the Author):

The manuscript by Tsarouchas et al. reports the important role that macrophages play in spinal cord axonal regeneration by producing pro-regenerative $\text{tnf-}\alpha$ and controlling the levels of the pro-inflammatory cytokine Il-1b . This work is of general interest and it brings a better understanding on the role of immune cell responses and inflammation after spinal cord injury in a successfully regenerating species. The studies presented are based on *in vivo* imaging in zebrafish in combination with the use of genetic and pharmacologic tools allowing for the manipulation of subpopulations of innate immune cells. Briefly, Tsarouchas et al. show that after complete spinal cord transection, macrophages, but not neutrophils or microglia, are necessary for functional spinal cord regeneration. All of the experimental timelines are clearly presented and the statistical analyses are done rigorously. The conclusions are generally supported by the data described in the paper, however, the

language used by the authors often overstates their results and should be more reflective of the findings they report in the paper. While the authors' work is generally rigorous and of great quality, there are points that could be improved regarding the use of CRISPR/cas9, and some changes are needed in the main text. I lay out some some major concerns below:

COMMENT 1. All throughout the article, the authors quantify the “percentage of larvae with bridged lesion site”, but do not examine the efficiency of the axonal bridging *per se* (e.g. quantification of the number of axons that fully regrew or measurement of the tubulin fluorescence intensity across the lesion site). Their method unfortunately does not take into account the variability in bridging among fish or among different experimental conditions and does not support their claim of “full regeneration” of the spinal cord. Adding a more rigorous assessment of axonal bridging would greatly improve the significance of the findings and may reveal more subtle changes in the regenerative capacity with regards to some of their manipulations.

RESPONSE: We agree with these helpful suggestions. We have performed additional experiments (new suppl. Fig. S2), edited out the term “full regeneration” and added more previously published data on the scoring method, which we inadvertently left out of the manuscript, to validate our quantification method:

-As suggested, we have now added new experiments in which we measure the thickness of the axonal bridge at the level of individual animals. These confirm an effect of dexamethasone on bridging of the injury site. We also observe this by our scoring method (new suppl. Fig. S2). In addition, these new data show a correlation of bridge thickness with touch-evoked swimming distance.

As a further experiment, we re-assessed axon bridging with this method for the rescue experiments of the *irf8* mutant with QVD. We do not see a difference between control and experimental group with our scoring method or with measuring the thickness of the bridge (added to the manuscript text on p. 10). Hence, scoring a

percentage of animals with bridged lesion sites is of sufficient sensitivity to indicate experimentally induced changes in axonal regeneration.

-We have previously shown (Wehner et al., 2017) that animals scored as non-bridged show significantly worse recovery of swimming distance. This is now mentioned at the beginning of the Results on p. 5.

-We have previously shown that re-lesion of the axonal bridge abolishes recovery of the swim distance parameter (Ohnmacht et al., 2016), underscoring the importance of the axonal bridge for recovery (now mentioned on p. 5 in the manuscript).

COMMENT 2. Tsarouchas et al. claim that macrophages are necessary and sufficient for full regeneration. I think this conclusion is overstated. Their work shows axonal bridging across the lesion site up to 5 days post lesion, but it by no mean shows full regeneration of the spinal cord. This study does not assess the presence of glial cells associated with axons post-bridging and does not show if lesioned axons eventually get remyelinated. The authors also do not present any satisfactory behavioral assay post-lesion, which should be demonstrated in order to claim a full regeneration of the spinal cord. Fig. 7E shows that inhibition of Il-1b “restores functional recovery” and “rescues impaired touch-evoked swimming” in lesioned *irf8*^{-/-}, however, the measurement of the swimming distance alone does not reflect the swimming pattern or speed. A more detailed assessment of swimming in these studies is important if claiming full functional recovery.

RESPONSE: We agree with this comment and performed additional experiments. While these provide more data on the regeneration process (see below), we still edited the text to avoid overstatements:

To better understand the degree of recovery of swimming parameters, we reanalysed the swim paths for the *irf8* rescue experiments in question for “mean velocity” and “meandering”. Both parameters were similarly impaired to “total distance moved” in *irf8* mutants and rescued to unlesioned control levels by YVAD treatment (new suppl. Fig.S10 I,J), supporting that some motor functions may be rescued by the treatment. However, we do not state that this is indicative of “full regeneration” or “full recovery” anymore.

-To gain insight into the reaction of astroglia-like cells, we determined bridging of astroglial-like processes and found that these are similarly affected to axons by dexamethasone treatment (new suppl. Fig. S2D).

-To emphasize the possible significance of myelinating glial cells, we now explicitly discuss the finding that axons become remyelinated, as indicated in Suppl. Fig. 1C. P. 22: “Although relatively late, axons become remyelinated, which may contribute to recovery of some swimming parameters after injury”.

COMMENT 3. For the purpose of validating the pharmacological inhibition of TNF- α , the authors generated somatic mutations in *tnf*- α by using the CRISPR/Cas9

genome editing tool and examined the percentage of injected larvae with a bridged lesion site after spinal cord transection (Fig.6 B and C). It is unclear however, whether all of the TNF- α gRNA injected larvae were individually genotyped and tested for somatic mutations, as the authors do not score the efficiency of the Cas9 cleavage. Although the RFLP analysis is a validated way to show that Cas9 produces cuts in the gene of interest, the authors do not provide any clonal analysis of the somatic mutations generated in the F0 larvae and do not prove that the cuts lead to indels. CRISPR/Cas9 injected embryos should be characterized more thoroughly and this requires more controls.

RESPONSE: We agree with this comment and performed more analyses, which show effectiveness of the gRNA:

- We now provide a list of indels we observed, many of them leading to predicted frame shifts or are in a conserved region, added as suppl. Table S1.
- We performed Western blots of 3-day-old embryos injected with *tnf-alpha* gRNA and found a reduction in protein level (new Suppl. Fig. S9B)
- We performed intensity measurements of *tnf-alpha* immunoreactivity in the injury site and found a reduction in fluorescence intensity (new Suppl. Fig.S9A)

COMMENT 4. In Fig.6C, Tsarouchas et al. show that axonal bridging is impaired after disruption of *tnf-alpha*. The authors do not rule out the possibility that TNF- α gRNA injected larvae may display a regenerative delay due to the injection of gRNA and cas9 mRNA, that often leads to developmental delays, and therefore should provide better controls. The authors only examined the injected larvae at 1dpl and 2dpl, while they could have examined the phenotype at 5dpl as they did with other mutants earlier in the paper (e.g. *irf8*^{-/-} in Fig.2B).

RESPONSE: We are grateful for these suggestions and have performed new experiments using a gRNA for GFP as an additional control and added a 5 dpl time-point as suggested (modified Fig. 5C). This confirms our findings and shows that the effect of *tnf-alpha* gRNA is long-lasting.

COMMENT 5. The authors show that TNF- α :GFP is mostly expressed in macrophages (95.7%) in the lesion site, as supported by the immunostainings for L-plastin, 4C4 and Mpx presented in Fig.6D. However, the authors do not mention in the manuscript that another source of TNF- α could potentially be axons themselves. At least one previous study has demonstrated that sensory axons express *tnfa*. The intrinsic expression of TNF- α in sensory axons may participate in creating a favorable environment for regeneration. For this reason, the use of the adjective "sufficient" seems inadequate.

RESPONSE: We agree and edited the manuscript to cite potential additional sources of Tnf-alpha (p. 13: "However, other cell types...") and rephrased to eliminate the term "sufficient".

COMMENT 6. Studies in adult zebrafish demonstrate the essential role that radial glia play in bridging spinal cord lesion sites and creating a scaffold for axonal regrowth. In larvae that are not deprived of radial glia, can the authors say that “immune system is necessary for and sufficient to promote axon regeneration”? Here again, the vocabulary may not be appropriate as radial glia may also be involved in their studies.

RESPONSE: We are grateful for these suggestions. We have eliminated the use of the term “sufficient” from the manuscript. We also performed new experiments to show that astroglial-like process that cross the injury site also depend on the immune response (see response to comment 2; Suppl. Fig.S2) and we discuss the relevance of glial processes in the discussion section on p. 22: “Astroglia-like process cross the injury site...”.

COMMENT 7. By doing qPCR analysis, the authors show a decrease in expression of the anti-inflammatory cytokines TGF- β 1a and TGF- β 3 during the late regenerative phase in *irf8*^{-/-}. The authors propose that TGF- β come from macrophages, but Fig.5C shows that the level of TGF- β 1a peaks 15 minutes post-lesion in *irf8*^{-/-}. This raises the question of the source of TGF- β other than macrophages.

RESPONSE: To determine the source of TGFs, we performed qRT-PCR on FACS sorted macrophages and rest tissue from about 500 animals. We detect both TGF- β 1a and TGF- β 3 in macrophages, but also in rest tissue (new suppl. Fig. S8B). We also performed ISH, showing wide-spread expression and strong label in cells around the injury site (new suppl. Fig. S8A). In addition, we cite literature that shows precedent for expression in non-immune cell types at the bottom of p. 11.

Reviewer #2 (Remarks to the Author):

The manuscript highlights the pivotal role of macrophages in axonal regeneration in the zebrafish.

The work is elegant, and the results are, on the whole, convincing.

Several comments:

COMMENT 1. The manuscript is somewhat overloaded with information. Some of the results are negative, and do not directly support the overall conclusion. Such results should be placed in the supplementary section.

RESPONSE: We are grateful for this helpful suggestion and put the study on the role of debris into the supplementary section, which hopefully enhances the flow of the manuscript.

COMMENT 2. Regarding experiments to determine the effect of macrophages on ECM composition- macrophages are the most obvious source of ECM degrading enzymes, and therefore the authors should consider testing a battery of enzymes that can potentially degrade the ECM

RESPONSE: We agree with this suggestion and performed a PCR survey of 21 annotated ECM related enzymes in the zebrafish genome. Five of these showed robustly lower expression in macrophage/microglial-less *irf8* mutants, suggesting potential changes in ECM modifications in the mutant. We have added the data as new suppl. Fig. S5.

COMMENT 3. The role of the phagocytic function of macrophages should be tested.

RESPONSE: We followed this interesting suggestion by incubating larvae in L-SOP, a pharmacological inhibitor of phagocytosis. This increased the abundance of cellular debris as expected, but had no influence on axon bridging, in line with our other results. This is now added as new suppl. Fig. S7.

COMMENT 4. Many of the literature citations are inaccurate.

RESPONSE: We are grateful to the reviewer for alerting us to this. We double-checked the citations and modified these also in relation to the request to discuss more literature on cytokine function (comment 6).

COMMENT 5. More insight is needed regarding the underlying role of TNF α .

RESPONSE: We agree with this point. To determine alterations of the immune response when Tnf-alpha function is compromised, we counted innate immune cells and measured levels of cytokines by qRT-PCR, showing that Il-1beta levels are roughly doubled and neutrophil numbers are increased when Tnf-alpha is inhibited. These data are now added to main Fig. 6C,D. Since neutrophils and Il-1beta are detrimental to axon crossing, we discuss this now as part of the mechanism by which lack of Tnf-alpha inhibits regeneration on p. 20: "Tnf- α may exert its positive role...".

COMMENT 6. A more comprehensive discussion should be devoted to the issue of how inflammatory associated cytokines have distinct and opposite roles in repair.

RESPONSE: We added a paragraph to the discussion which discussed the differences between TNF-alpha and Il-1beta on p. 20: "Different functions for the two pro-inflammatory cytokines have been reported.."

COMMENT 7. The Scheme of the model is not self-explanatory; the temporal issue should be emphasized.

RESPONSE: We agree with this comment and split the model into two panels reflecting early and late immune system actions (modified Fig. 10).

Reviewer #3 (Remarks to the Author):

This manuscript by Tsarouchas et al describes their results from a comprehensive analysis of the role of innate immune responses in axon regeneration in zebrafish. An initial finding is that macrophage-deficient mice (*irf8* mutants) exhibit defects in axon regeneration, and this appears to be largely mediated by peripheral macrophages, instead of microglia. An important underlying regenerative mechanism

is the prolonged inflammation after injury, with elevated levels of IL1beta and TNFalpha. Their further studies showed that these factors differ in their role in impacting axon regeneration, with TNFalpha shown to promote, and IL1beta shown to be binary (initially promoting and later inhibiting). Most conclusions were supported by more than one line of evidence and are largely convincing to me. I think that these results together provide an interesting and comprehensive picture of the involvements of innate immunity in the processes of spinal cord injury and axon regeneration in the fish model. There are a few issues to be fixed prior to publication.

COMMENT: Some of manipulations need better additional verification. For example, several methods were used to block TNFalpha, but to what extents TNFalpha's expression was reduced was not assessed.

RESPONSE: We have now determined that protein levels after Tnf-alpha gRNA injection are clearly reduced by about half by Western Blot and whole-mount immunohistochemistry (new suppl. Fig. S9A,B); not all of the still detectable protein may be functional, as even in frame mutations are in a highly conserved domain of the protein (Savan et al., 2005); see also response to comment 3 of reviewer #1

COMMENT: The authors indicated a distinct role of peripheral macrophages and microglia. The question is whether this is a quantitative or a qualitative difference. Are there any differences in terms of their gene expression relevant to this context?

RESPONSE: We agree that this is an interesting point. We re-assessed expression of tnf-alpha and il-1beta in these cell types and find that il-1beta is expressed by both cell types at similar ratios (new Fig. S12A), but that Tnf-alpha is mostly expressed by macrophages. This underscores the important role of peripheral macrophages for axon bridging. The data is added as new Fig. 6C and discussed on P. 20." They also differ in cell type of origin. ...".

COMMENT: The relationship between axon bridge and swimming behavioral recovery: From the text, axon bridge formation and swimming recovery appear with a similar temporal window. Assuming that regenerated axons need to make synapses, it is unclear whether behavioral recovery is entirely due to regenerating axons. What portions of axon regeneration are required for swimming behavioral? Because multiple manipulations affect axon regeneration to different extents in this manuscript, it will be useful for the authors to analyze whether there is a clear correlation.

RESPONSE: This is a helpful suggestion. We performed a regression analysis of thickness of the axonal bridge of the lesion site and behavioural recovery, which shows that these are positively correlated (new suppl. Fig. S2A-C). We now also cite previous evidence that axon bridging is important for functional recovery; see response to COMMENT 1 of reviewer #1.

COMMENT: Finally, this manuscript could be improved by additional editing. There are a number of errors in this current manuscript. For example, the legend for Fig. 1E and I are missing. This is an important missing point, because it is unclear how

axonal bridges were assessed in this case and whether it is the same method as other figures.

RESPONSE: We apologize for this oversight. This is now corrected. We used the scoring method for axonal bridging throughout the manuscript and now include additional experiments and information to validate the method (see response to reviewer #1; comment 1).

Savan, R., Kono, T., Igawa, D., Sakai, M., 2005. A novel tumor necrosis factor (TNF) gene present in tandem with the TNF-alpha gene on the same chromosome in teleosts. *Immunogenetics* 57, 140-150.

Reviewers' Comments:

Reviewer #1:

Remarks to the Author:

Tsarouchas et al. have done elegant and novel work describing the role of the innate immune responses in axon regeneration after spinal cord transection in zebrafish. The study depicts the pro-regenerative role of macrophage TNF- α in axon regeneration and behavioral recovery.

This manuscript presented by Tsarouchas et al. has considerably improved both in terms of the text and at the experimental level. The study is thorough and the conclusions are suitably supported by the results and are convincing.

The authors clarified important aspects of the genetic manipulations using the CRISPR/Cas9 technology and now provide a rigorous characterization of the TNF- α F0 larvae. New data, as well as citations referring to previous findings, have been added to the text in order to better support the scoring method used throughout the article.

A previous concerning caveat was that the authors quantified the percentage of larvae with axonal bridging to the detriment of the analysis of individual animals. New experiments assessing the thickness of the axonal bridge at the level of individual larvae have been added to the article and now make the study more complete.

The authors have broadened their study by investigating glial cells in axon regeneration, and hypothesize that remyelination might participate to some extent in swimming recovery.

Lastly, the authors now present a nice regression analysis that positively correlates thickness of the axonal bridge of the lesion site and behavioral recovery.

For these reasons, I feel that this manuscript should be published in Nature Communications.

Reviewer #2:

Remarks to the Author:

The authors addressed most comments raised by the reviewers.

The only issue that was not satisfactorily addressed is the discussion, in connection with previous works. While the authors emphasized that the regeneration in this model is dependent on macrophages rather than microglia, they failed to cite and discuss works, published by several groups in high profile journals, which showed this phenomenon in rodents, in models of spinal cord and optic nerve .

Reviewer #3:

Remarks to the Author:

In this revised manuscript, the authors have addressed all of my concerns raised during previous review. I now support the publication of this manuscript.

RESPONSE TO REVIEWERS

Reviewer #1 (Remarks to the Author):

Tsarouchas et al. have done elegant and novel work describing the role of the innate immune responses in axon regeneration after spinal cord transection in zebrafish. The study depicts the pro-regenerative role of macrophage TNF- α in axon regeneration and behavioral recovery.

This manuscript presented by Tsarouchas et al. has considerably improved both in terms of the text and at the experimental level. The study is thorough and the conclusions are suitably supported by the results and are convincing.

The authors clarified important aspects of the genetic manipulations using the CRISPR/Cas9 technology and now provide a rigorous characterization of the TNF- α F0 larvae. New data, as well as citations referring to previous findings, have been added to the text in order to better support the scoring method used throughout the article.

A previous concern was that the authors quantified the percentage of larvae with axonal bridging to the detriment of the analysis of individual animals. New experiments assessing the thickness of the axonal bridge at the level of individual larvae have been added to the article and now make the study more complete.

The authors have broadened their study by investigating glial cells in axon regeneration, and hypothesize that remyelination might participate to some extent in swimming recovery.

Lastly, the authors now present a nice regression analysis that positively correlates thickness of the axonal bridge of the lesion site and behavioral recovery.

For these reasons, I feel that this manuscript should be published in Nature Communications.

RESPONSE: We thank the reviewer for previous helpful suggestions and this detailed response to our changes. No further changes are necessary

Reviewer #2 (Remarks to the Author):

The authors addressed most comments raised by the reviewers.

The only issue that was not satisfactorily addressed is the discussion, in connection with previous works. While the authors emphasized that the regeneration in this model is dependent on macrophages rather than microglia, they failed to cite and discuss works, published by several groups in high profile journals, which showed this phenomenon in rodents, in models of spinal cord and optic nerve.

RESPONSE: We thank this reviewer for helpful suggestions. We added two more citations to the manuscript to address this point, which together with the extensive discussion of the (macrophage-derived) cytokines hopefully gives a fair reflection of the literature. We apologize for any work that is not represented due to space restrictions.

Reviewer #3 (Remarks to the Author):

In this revised manuscript, the authors have addressed all of my concerns raised during previous review. I now support the publication of this manuscript.

RESPONSE: We thank this reviewer for their previously helpful suggestions. No further changes are necessary.